

**A weather regime characterisation of winter biomass aerosol transport from southern Africa**
Marco Gaetani (1,2); Benjamin Pohl (3); Maria del Carmen Alvarez Castro (4,5); Cyrille Flamant (6);
Paola Formenti (1)
(1) Institut Pierre Simon Laplace, Laboratoire Interuniversitaire des Systèmes Atmosphériques, UMR
CNRS 7583, Université Paris-Est Créteil, Université de Paris, Institut Pierre Simon Laplace, Créteil,
France
(2) Scuola Universitaria Superiore IUSS, Pavia, Italia
(3) CRC/Biogéosciences, UMR6282 CNRS / Université de Bourgogne Franche-Comté, Dijon, France
(4) University Pablo de Olavide, Seville, Spain
(5) CMCC, Bologna, Italy
(6) Institut Pierre Simon Laplace, Laboratoire Atmosphères, Milieux, Observations Spatiales, UMR
CNRS 8190, Sorbonne Université, Université Versailles Saint Quentin, Paris, France
Contact: marco.gaetani@iusspavia.it




**Abstract**
During austral winter, a compact low cloud deck over South Atlantic contrasts with clear sky over
southern Africa, where forest fires triggered by dry conditions emit large amount of biomass burning
aerosols (BBA) in the free troposphere. Most of the BBA burden crosses South Atlantic embedded in
the tropical easterly flow. However, midlatitude synoptic disturbances can deflect part of the aerosol
from the main transport path towards southern extratropics.
In this study, a characterisation of the synoptic variability controlling the spatial distribution of BBA
in southern Africa and South Atlantic during austral winter (August to October) is presented. By
analysing atmospheric circulation data from reanalysis products, a 6-class weather regime (WR)
classification of the region is constructed. The classification reveals that the synoptic variability is
composed by four WRs representing disturbances travelling at midlatitudes, and two WRs
accounting for pressure anomalies in the South Atlantic. The WR classification is then successfully
used to characterise the aerosol spatial distribution in the region in the period 2003-2017, in both
reanalysis products and station data. Results show that the BBA transport towards southern
extratropics is controlled by weather regimes associated with midlatitude synoptic disturbances. In
particular, depending on the relative position of the pressure anomalies along the midlatitude
westerly flow, the BBA transport is deflected from the main tropical route towards southern Africa
or the South Atlantic.
This paper presents the first objective classification of the winter synoptic circulation over South
Atlantic and southern Africa. The classification shows skills in characterising the BBA transport,
indicating the potential for using it as a diagnostic/predictive tool for aerosol dynamics, which is a
key component for the full understanding and modelling of the complex radiation-aerosol-cloud
interactions controlling the atmospheric radiative budget in the region.





## 1. Introduction

Natural and anthropogenic tropospheric aerosols are fundamental ingredients of the climate system. They influence the radiative properties of the atmosphere by deflecting and absorbing radiation (direct effect) and the cloud formation and properties by absorption (semi-direct effect) as well as by acting as cloud condensation nuclei (indirect effect). As a consequence, aerosols can influence on the atmospheric synoptic and large-scale dynamics (Bellouin et al., 2020).

Africa is the Earth's largest source of biomass burning aerosol (BBA; e.g. van der Werf et al., 2010, 2017), The transport of BBA, originated from central Africa and embedded in the tropical mid-tropospheric easterly flow, occurs mostly above the Atlantic Ocean (Fig. 1a), a prominent feature during austral winter (June to October; Fig. 1b) between the Equator and 20°S, when dry conditions in central Africa favour the development of forest fires (Horowitz et al., 2017). However, extratropical rivers of smoke are also observed to extend to 30-40°S between August and October (Fig. 1b). The definition 'river of smoke' refers to the sharply defined boundaries of the smoke plume, which can be several hundred kilometres wide and flow over a few thousands kilometres above southern Africa towards Southern and Indian Oceans (McMillan et al., 2003; Swap et al., 2003). Depending on the transport path (e.g. either above the continent or recirculated above the ocean), physical and chemical properties of the BBA may change (Abel et al., 2003; Eck et al., 2003; Formenti et al., 2003; Haywood et al., 2003; Pistone et al., 2019; Wu et al., 2020). The characterisation of BBA transport in terms of synoptic atmospheric circulation is therefore one of the key elements to shed light the already complex picture of the radiation-aerosol-cloud interactions (Adebiyi and Zuidema, 2018; Formenti et al., 2019; Haywood et al., 2003, 2021; Lindesay et al., 1996; Mallet et al., 2020; Redemann et al., 2021; Swap et al., 2003; Zuidema et al., 2016). Additionally, the extent and direction of BBA transport may condition the atmospheric remote supply of nutrients and pollutants to South Atlantic, the Southern ocean, and the Indian Ocean, as well as to the Antarctica (Baker et al., 2010; Barkley et al., 2019; Gao et al., 2003, 2020; Swap et al., 1996; Wai et al., 2014).

Understanding the role of the radiation-aerosol-cloud interaction in controlling the atmospheric radiative budget and, consequently, climate dynamics is a key aspect for the improvement of climate modelling. Indeed, even state-of-the art climate models still struggle in reliably representing the atmospheric radiative forcing, due to inaccurate parametrizations of the radiation-aerosol-cloud interaction (Mallet et al., 2020; Stier et al., 2013; Tang et al., 2019). This is particularly relevant in the South Atlantic, where the incomplete knowledge of the smoke-cloud regime generates large discrepancies in the modelling of radiative forcing and sea surface temperature (SST) in the region, eventually affecting climate simulations at regional and global scale (Zuidema et al., 2016). While a conceptual understanding of the meteorological conditions determining the transport of aerosols



and pollutants at the subcontinental scale exists (Diab et al., 1996; Garstang et al., 1996; Tyson,
1997), the large-scale drivers of the aerosol spatial distribution in the region are still not understood
and an objective synoptic characterisation of the wintertime BBA transport is still missing to date. In
particular, an objective synoptic characterisation of the wintertime BBA transport in the region is still
missing to date. Indeed, synoptic circulation in the southern Africa/South Atlantic sector is discussed
in literature mainly in relationship with convection and precipitation during austral summer (e.g.
Crétat et al., 2019; Dieppois et al., 2016; Fauchereau et al., 2009; Macron et al., 2014; Pohl et al.,
2018; Vigaud et al., 2012).
In this paper, an objective weather regime (WR) classification of the winter atmospheric circulation
in the southern Africa/South Atlantic sector is presented for the first time, and used to characterise
the BBA transport in the region. In particular, the study focuses on the characterisation of the
southward deflection of BBA from the mean tropical easterly flow from August to October (ASO) in
the period 2003-2017. Atmospheric circulation data from a reanalysis product are first used to
classify the synoptic circulation patterns. Then, the classification is used to characterise the BBA
transport anomalies in reanalysis data and in situ observations in the region.
The paper is organised as follows: in Section 2, data and methods used in the analysis are presented;
in Section 3, the WR classification is presented and the synoptic characterisation of BBA anomalies is
discussed; conclusions and perspectives are summarised in Section 4.
**2. Data and Methods**
**2.1. Reanalysis and observations**
The atmospheric circulation and the spatial distribution and optical properties of the BBA over
southern Africa and the South Atlantic in ASO 2003-2017 are analysed using data from the
Copernicus Atmospheric Monitoring Service reanalysis product (CAMS; Flemming et al., 2017) at 6h
time steps and 0.75° horizontal resolution. Daily values are obtained at each grid point as the
average of 6h time steps, and daily anomalies are computed by removing the low frequency
component of the time series, estimated by computing monthly means from daily data and
interpolating them to daily time steps using a cubic spline interpolation. The limited coverage of the
CAMS reanalysis (15 years) does not allow a robust definition of the climatological seasonal cycle,
which would be too dependent on the interannual variability. Therefore, in order to isolate the
synoptic variability alone, the definition of a low frequency component, accounting for the seasonal
cycle and the interannual variability, is preferred. The BBA emission is estimated by the organic
matter mixing ratio at 10m, the BBA transport is estimated as the product of organic matter mixing



ratio and wind at 700 hPa, and the aerosol spatial distribution is represented by the AOD at 550 nm
(Fig. 2a).
Global reconstructions of the observed sea surface temperature (SST) are used to investigate the
teleconnections controlling the synoptic variability. Data are extracted from the Met Office Hadley
Centre HadISST dataset (Rayner et al., 2003), available from 1871 at monthly time scale and 1°
horizontal resolution, and from the NOAA Extended Reconstructed Sea Surface Temperature (ERSST)
Version 5 dataset (Huang et al., 2017), available from 1854 at monthly time scale and 2° horizontal
resolution.
Observed daily values of the aerosol optical depth (AOD) at 500 nm from AERONET stations
(https://aeronet.gsfc.nasa.gov/) are used to validate the synoptic characterisation performed on
CAMS data. Stations are selected among the ones with at least 2 years of level 2 data obtained from
the Version 3 Direct Sun algorithm (Giles et al., 2018). Stations are selected outside the source
region in Tropical Africa, namely south of 20°S and west of 10°E (Fig. 2a, Table 1), in order to focus
on BBA transport only, not being influenced by the BBA emission which is assumed not to be directly
related to synoptic conditions. Among the available stations, St. Helena (15.9°S, 5.7°W) and Wits
University (26.2°S, 28.0°E) are not included because of the limited coverage (less than 100
observations during the study period). Moreover, the stations closer to the greater Johannesburg
and Pretoria urban areas (namely, Durban UKZN (29.8°S, 30.9°E), Pretoria CSIR-DPSS (25.8°S, 28.3°E)
and Skukuza (25.0°S, 31.2°E)) are not included, because too affected by the proximity with urban
sources (Fig. 2a). For each station, daily AOD anomalies are computed by removing the seasonal
cycle at the station. However, the sparseness of the AERONET observations makes difficult the
definition of a daily seasonal cycle. Therefore, CAMS AOD at 550 nm is selected in an area defined by
the grid point the closest to the station coordinates and the adjacent grid points, and averaged to
estimate the daily seasonal cycle of the AOD at 500 nm. Empirical evidence shows that a quadratic
relationship exists between the natural logarithm of AOD and wavelength (Eck et al., 1999).
However, at such close wavelengths the relationship can be assumed as linear, and the relationship
between the natural logarithm of AOD at 500 and 550 nm can be modelled as follows:
$$\log AOD_{500nm} = a \log AOD_{550nm} + b.$$
At each AERONET station, the logarithm of observed and CAMS AOD well correlates during ASO
2003-2017 (correlations coefficients lie between 0.71 and 0.90, all significant at 99% level of
confidence, see Fig. 3). Therefore, the daily seasonal cycle of the observed AOD is estimated by
means of a linear regression onto the CAMS seasonal cycle. In order to minimise the effect of
possible large discrepancies between AERONET and CAMS data, the difference between AERONET



and CAMS AOD is computed and the values in the lowest and highest 5% are discarded before to
perform the linear regression (the coefficients used in the regression model at each station are
displayed in Fig. 3).
**2.2. Weather regime classification**
The WR classification is performed on the geopotential height at 700 hPa, which is the level where
BBA transport is maximal, in the domain [20°W-40°E, Eq-40°S] (see Fig. 2bc). The selection of the
domain is made to include the main BBA transport routes in the tropical belt and towards the
extratropics. However, during the dry season the synoptic variability in the tropics is reduced in
comparison with the extratropics (Baldwin, 2001). Therefore, the southern border of the domain is
set to 40°S to not let the dominant midlatitude modes to mask variability in the tropical belt. The
atmospheric circulation is first characterised by isolating the main modes of variability represented
by the empirical orthogonal functions (EOFs) of the geopotential height derived by a principal
component analysis (PCA). Each mode is represented by a spatial anomaly pattern and a
standardized time series (namely, the principal components, PCs) accounting for the amplitude of
the anomaly pattern (for more details on PCA, see Storch and Zwiers, 1999). The first 4 EOFs,
accounting for 80% of total variability (Fig. S1), are used to classify the WRs by means of a k-means
algorithm, using k = [2, 10] (Michelangeli et al., 1995). For each k, the classification is performed 100
times, to ensure reproducibility of the results. A red-noise test is performed to assess the
significance of the class partition (Michelangeli et al., 1995), resulting in 6 and 7 classes (Fig. S3). In
this study, the 6 class partition is used. The synoptic characterisation is also performed by using the
7 class partition, and is illustrated in Section S2.
**2.3. Synoptic characterisation**
The WR classification is used to characterise the observed AOD data from the AERONET stations in
the region (Table 1). Two approaches are used:
1) Daily AOD anomalies are linked to the corresponding WR and grouped, and statistical differences
among groups are investigated (circulation-to-environment approach, C2E). The significance of the
C2E characterisation is assessed by a one-way analysis of variance (ANOVA) with the null hypothesis
that the distributions associated with each WR are derived from populations with the same mean.
Furthermore, for each WR the significance of the associated AOD anomalies with the respect of the
full sample is assessed by a non-parametric Kolmogorov-Smirnov (KS) test.
2) Daily AOD anomalies are divided into quartiles, and the changes in the WR occurrences within
each quartile are studied (environment-to-circulation approach, E2C). The significance of the E2C
characterisation is assessed by computing the Chi-squared statistics for each quartile, with the null



hypothesis that the associated WR frequencies are derived from the same distribution of the full
sample. The Chi-squared statistics is tested against the critical value for 5 degrees of freedom and at
the 95% level of confidence. The degrees of freedom are estimated as the number of the
observation categories (6 WRs) minus the parameters of the distribution to be fitted (the mean WR
occurrence, i.e. 1).

## 3. Results

### 3.1. Synoptic characterisation of the regional atmospheric variabiliy

The mean atmospheric conditions over South Atlantic and southern Africa in ASO are illustrated in
Fig. 2a. The atmospheric circulation at 700 hPa is characterised by a continental high centred at 25°S
over southern Africa and extending over eastern South Atlantic, and a subtropical trough west of
South Africa deflecting southward the midlatitude westerly flow. Massive quantities of BBA are
emitted from tropical southern Africa, and are driven westward over South Atlantic by the easterly
trade winds, while the anticyclonic gyre associated with the continental high recirculates the BBA
towards South Africa along the Namibian coast. This recirculation merges with smaller BBA amounts
emitted from sources located in South Africa in the urban area of Johannesburg and Pretoria, to be
eventually transported eastward to the Indian Ocean embedded in the westerly flow.
The WR classification shows two synoptic patterns (WR2 and 6) accounting for the oscillation of the
pressure filed in the South Atlantic and four synoptic patterns (WR1, 3, 4 and 5) accounting for
midlatitude pressure anomalies (Fig. 4). These four WRs represent the fingerprint of propagative
disturbances travelling along the midlatitude mean westerly flow with wave number 8-12, as
demonstrated by the EOF analysis (see Section S1). The synoptic variability is dominated by the WR2,
which occurs at a frequency of 22.3% and is characterised by a high pressure anomaly in the South
Atlantic accompanied by a reinforcement of the midlatitude westerlies (Fig. 4a). Its symmetric
counterpart is represented by WR6, which occurs at a frequency of 17.7% and is characterised by a
low pressure anomaly and a weaker westerly flow in the midlatitudes (Fig. 4f). WR2 occurs mainly in
September-October, while WR6 does not show a clear seasonality (Fig. 5). The analysis of the
transitions from a WR into the others reveals that WR2 and 6 are dominated by persistence (Table
2). The remaining 60% of the synoptic variability in the region is characterised by eastward travelling
disturbances of the westerly flow, represented by WR1, 3, 4 and 5 (Fig. 4). WR1 and 3 occur more
frequently in August-September, while WR4 and 5 are more frequent in October (Fig. 5). In this case,
the analysis of the transition rates shows similar persistence ratios (from 0.39 to 0.46) and high rates
for preferred transitions (WR1 into 5, WR3 into 4, WR4 into 1, WR5 into 3, see Table 2), pointing out
the propagative character of these WRs.



At the global scale, the variability of the atmospheric circulation south of 20°S is dominated by the
southern annular mode (SAM), which consists of out-of-phase surface pressure and geopotential
height anomalies between the Antarctic region and the southern midlatitudes, resulting in the
modulation of the location and intensity of the westerly wind belt (Baldwin, 2001; Limpasuvan and
Hartmann, 1999). Pohl and Fauchereau (2012) characterised the synoptic variability of the SAM in
terms of WR, identifying 4 main variability modes in the southern midlatitudes, three of them
associated with circulation patterns characterised by stationary wave-number 4. The persistent
character of WR2 and 6 indicate a possible connection with the synoptic variability of the SAM. The
relationship between the WR occurrence and the SAM daily index is investigated using both the C2E
and the E2C approach. WR6 shows a statistically significant association with positive SAM phases
(not shown), coherently with expected weaker westerlies at midlatitudes (see Fig. 4f). However, the
WR2-SAM relationship results statistically weaker, and no relationship at all is found with WR1, 3, 4
and 5 (not shown).
The WRs describing propagative disturbances at midlatitudes (WR1, 3, 4 and 5) are characterised by
the longitudinal displacement of high-low pressure anomalies modulating the meridional circulation,
which in turn drives the poleward BBA transport above the South Atlantic and southern Africa. In
particular, WR3 favours the recirculation of BBA from the ocean towards Namibia and South Africa,
leading to significant positive AOD anomalies above all the continental stations (Fig. 4c), while WR5
pushes the BBA recirculation above the South Atlantic and inhibits the BBA transport towards the
Indian Ocean (Fig. 4e). Conversely, WR1 and 4 are associated with a weaker BBA transport above
Namibia and South Africa, leading to significant negative anomalies above the continental stations,
and larger transport towards the Indian Ocean (Fig. 4ad). BBA transport along the Atlantic coast of
Namibia and South Africa is also anomalously high during WR6, which is characterised by a low
pressure anomaly in the South Atlantic inhibiting the transport towards subtropical South Atlantic,
and leading to significant negative anomalies above Ascension Island, and favouring a poleward
route driving anomalous BBA concentrations above the continental stations (Fig. 4f). WR2,
characterised by a high pressure anomaly in the South Atlantic strengthening the easterly flow in the
Tropics, is the only WR associated with a reinforcement of the main BBA transport route in the
tropical South Atlantic, and positive AOD anomalies only affect the Ascension Island station (Fig. 4b).
**3.2. Synoptic characterisation of aerosol optical depth in-situ observations**
The robustness of the synoptic characterisation of the BBA transport obtained from the CAMS data
is assessed by linking the WR classification to the observed AOD from AERONET stations in the
region (Table 1).



The C2E characterisation of the AOD observations is presented in Fig. 6; the associated statistical
analysis is summarised in Table 3. AOD anomalies above Ascension Island during WR1-5 are evenly
distributed between positive and negative values, while AOD anomalies during WR6 show a
preference for negative values (Fig. 6a). The significance of this characterisation is confirmed by the
ANOVA with a level of confidence higher than 99%. Just south of the source region in Bonanza,
significant positive anomalies are observed during WR4 (Fig. 6b). However, the statistical significance
of this characterisation only reaches 93%. AOD variability at central Namibia stations (Gobabeb,
Henties Bay and HESS) is dominated by WR1, leading to significant negative anomalies (Fig. 6c-e). In
Gobabeb and HESS, significant positive anomalies are observed during WR6 (Fig. 6c) and WR3 (Fig.
6e), respectively. Negative anomalies are also observed in Gobabeb and Henties Bay in association
with WR4, however these anomalies are poorly significant (Fig. 6c,d). The ANOVA supports this
characterisation, indicating that the null hypothesis can be rejected with a level of confidence higher
than 99%. Similarly, the continental station in Upington shows significant negative anomalies during
WR1 and 4, and significant positive anomalies during WR3 (Fig. 6g), and the ANOVA indicates the
rejection of the null hypothesis with 99% level of confidence. In South Africa, the southernmost
station in Simon's Town does not show significant anomalies in association with any WR, and the
ANOVA confirms that the WR classification is not able to characterise the AOD variability (p=0.09).
The C2E characterisation performed using observed AOD data confirms the relationship between the
WRs associated with midlatitude disturbances (WR1, 3 and 4) and the BBA transport above the
AERONET continental stations, and between WR6 and the BBA transport above Ascension Island, as
shown by the CAMS data (cf. Fig. 4). The comparison with the synoptic characterisation performed
using a WR classification with 7 clusters highlights that the latter is less robust, showing poorer
ANOVA performances. Moreover, the additional WR, accounting for a strengthening of the
continental high, does not provide further characterisation of the AOD anomalies (see Section S2 for
details).
The E2C characterisation of the BB AOD station data is presented in Fig. 7; the associated statistical
analysis is summarised in Table 4.  AOD anomalies are divided in quartiles, with quartiles from 1st to
4th representing anomalies from the largest negative to the largest positive, and the relative change
in WR occurrence is displayed for each quartile. In Ascension Island, the 3rd quartile is characterised
by a significant change in the WR frequency the distribution, with increased occurrence of WR4 (Fig.
6a). The Bonanza station does not show any significant change in the WR occurrence (Fig. 6b). In
central Namibia stations (Gobabeb, Henties Bay and HESS; Fig. 6c-e), positive AOD anomalies are
associated with significantly more frequent WR6. In HESS, positive anomalies are also associated
with significant changes in the occurrence of WR2-4. Significant changes in the WR distribution are


observed for the 1st quartile in Gobabeb (increased occurrence of WR1, 4 and 5) and HESS (more
frequent WR1 and 6). Similarly, the South African stations in Upington and Simon's Town show
positive AOD anomalies associated with more frequent WR3, 5 and 6, and negative anomalies
associated with more frequent WR1, 4 and 6 (Fig. 6f,g). The E2C characterisation confirms the
importance of the midlatitude disturbances (WR1, 3 and 4) in controlling the AOD anomalies at the
AERONET continental stations, in particular by driving the largest anomalies (1st and 4th quartiles).
However, this approach shows some inconsistencies: WR4, which is characterised by a southerly
anomaly in the BBA transport along the Atlantic coast (Fig. 4d), is associated with positive AOD
anomalies in HESS instead; similarly WR6, characterised by a northerly BBA transport anomaly along
the coast (Fig. 4f), is associated with both positive and negative anomalies in HESS and Upington.
The origin of this ambiguities is likely due to the location of these stations at the margin of the BBA
transport path associated with the WR circulation patterns, making them highly sensitive to the
variability of the circulation around the centroid. The comparison with the synoptic characterisation
performed using a 7 cluster WR classification highlights the same ambiguities when the AOD
anomalies in the continental stations are associated with the WR describing a low pressure anomaly
in the South Atlantic (see Section S2 for details).
**3.3. Interannual variability**
The WR frequency in ASO is also analysed at the interannual time scale. All WRs show similar
interannual variability in the frequency of occurrence (2-4% standard deviation), with the exception
of WR2 showing the larger interannual variability (6% standard deviation) (Fig. 8). No trend is found
in the WR occurrence (a Mann-Kendall test is performed at 95% level of confidence), not surprisingly
due to the short time coverage of the reanalysis. Possible teleconnections controlling the WR
interannual variability are analysed by computing the linear correlation between the WR frequency
and the SST variability at the global scale (Fig. 9). WR1, 3, 4 and 5 do not show significant correlation
patterns at the global scale, with the exception of localised SST anomalies in the South Atlantic
(WR1, 4 and 5) and Warm Pool (WR1). Conversely, WR2 and 6 show a strong relationship with El
Niño/Southern Oscillation (ENSO)-like patterns. In particular, WR2 occurrence is associated with La
Niña conditions, while WR6 is associated with El Niño conditions. The linkage with La Niña conditions
can also explain the larger interannual variability of WR2 during the analysed period, mainly due to
the peak in 2010 associated with a strong La Niña event (Boening et al., 2012), and the minimum in
2015, associated with an extreme El Niño event (Hu and Fedorov, 2017). The analysis of the WR-SST
correlations performed by using NOAA ERSST data show similar teleconnection patterns (see Section
S3). Differently from the WRs associated with travelling disturbances, WR2 and 6 represent a sort of
stationary South Atlantic oscillatory pattern, which might interact at the synoptic time scale with



Rossby-wave patterns from the equatorial Pacific during ENSO active phases (e.g. Hoskins and
Ambrizzi, 1993). The teleconnection mechanisms are explored by computing the correlation
between the WR occurrence and the global geopotential at 200 hPa (Fig. 10), the level where
teleconnection signals are the strongest. Wave patterns connecting the Pacific to South Atlantic are
found for both the WR2 and 6, though significance for WR6 is weak. A similar modulation by the
ENSO of synoptic regimes in the Southern Hemisphere is also reported during austral summer by
Fauchereau et al. (2009) and Pohl et al. (2018). The correlation between the occurrence of WR1, 3, 4
and 6 and the global geopotential does not show organised patterns at mid-to-high latitudes (Fig.
10). The comparison with the 7 cluster WR classification shows similar results, however the
teleconnection patterns are less evident (see Section S2).
The impact of the WR interannual variability on the BBA transport is assessed by computing the
linear correlation with the CAMS organic matter mixing ratio and the BBA transport at 700 hPa (Fig.
10). The WR interannual variability affects the mid-tropospheric circulation in the subtropics,
modulating the BBA transport on both the zonal and the meridional direction. However, the
correlation analysis reveals that the WR variability has weak impact on the BBA transport at the
interannual time scale, only controlling limited areas in the Tropical Atlantic (WR1), South Atlantic
(WR2) and southern Africa (WR4). However, the short time coverage and sparseness of the
AERONET observations does not allow validation of the impact of the WR characterisation on the
interannual time scale. The comparison with the 7 cluster WR classification does not show major
differences (see Section S2). In particular, the additional WR, accounting for a strengthening of the
continental high, shows no significant impact on the BBA transport (see Fig. S11a).
**4. Conclusions**
In this paper, the first objective classification of the synoptic circulation over South Atlantic and
southern Africa during the dry season is presented. By using atmospheric circulation data from a
reanalysis product, a robust classification with 6 WRs is defined for August-to-October in the period
2003-2017. Four WRs (WR1, 3, 4 and 5) represent the fingerprint of midlatitude propagative
disturbances, while two WRs (WR2 and 6) are characterised by persistence and represent the
oscillation of the pressure field in the South Atlantic. In particular, WR2 is associated with a
reinforced South Atlantic anticyclone, and is the dominant WR during the dry season. The
stationarity of the WR2/6 system suggests a connection with the synoptic variability of the SAM,
which is also consistent with the South Atlantic Oscillation pattern firstly identified by Chen (2014).
At the interannual time scale, the occurrence of persistent WR2 and 6 also shows a strong
connection with the El Niño/Southern Oscillation through a tropical-extratropical Rossby wave
pattern.





The synoptic classification is used to characterise the transport of BBA from equatorial Africa, which
dominates the aerosol atmospheric content in the region during the dry season. By analysing
reanalysis data, it is found that WR2 and 6 modulate the easterly transport from tropical Africa
sources, which is the main climatological transport route. The synoptic characterisation also shows
that midlatitude propagative disturbances modulate the BBA transport from equatorial Africa,
elucidating the mechanism responsible for the BBA transport to the extratropics, which is peculiar in
this period of the year. Specifically, WR3 drives enhanced transport above the continent, while WR1
inhibits the transport; WR5 drives the BBA recirculation over the ocean, which is inhibited by WR4.
The BBA transport characterisation is also tested by using AOD observations from AERONET stations,
which show a good degree of consistency with the results based on reanalysis data. However,
limited data availability in most of the stations prevents a robust statistical validation of the synoptic
characterisation of observations at the regional scale. Results show that the occurrence of WR1 and
4 inhibits the BBA transport towards the continental stations (Gobabeb, Henties Bay, HESS and
Upington), while WR3 favours the transport above the same locations. Along the Atlantic route, the
occurrence of WR6 limits the BBA transport towards Ascension Island. In-situ observations in
Bonanza and Simon's Town are not well characterised by the WR classification. The former likely
because of its proximity to the source region, where emission is not strongly affected by the synoptic
atmospheric circulation, the latter possibly because of the poor data coverage. WR then clustering
shows to be a valuable tool in discriminating aerosol transport and concentrations over South
Atlantic and southern Africa at the short timescales (day-to-day and synoptic variability). However,
the characterisation of the AOD variability at the interannual time scale shows limited performance,
probably due to the shortness of the time period considered in the analysis. Indeed, within a 15 year
time range, a large fraction of the variance is associated with daily weather patterns instead of
changes from one season to another. This gap can be filled by analysing longer coverage reanalysis
products.
A 7-class WR classification is also tested for the characterisation of the synoptic variability of the BBA
transport. However, this classification does not improve the performance of the 6 cluster WR
classification, showing overall poorer statistics and not correcting some ambiguities found in the E2C
characterisation of the continental AERONET stations.
This paper provides new insights in the understanding of the synoptic circulation in South Atlantic
and southern Africa, by characterising for the first time the dry season circulation and the associated
rivers of smoke. The characterisation of the transport routes in the region is crucial to support the
characterisation of the physical and chemical properties of the BBA, and model the associated
impact on clouds and radiation. The WR characterisation is also a valuable resource to develop



predictive tools for the BBA spatial distribution in the region. In particular, by using reliable long
coverage reanalysis products a classification for past decades can be built, and the BBA spatial
distribution can be reconstructed where observations are not available. Furthermore, the WR
characterisation can be used in climate model projections to estimate the future evolution of the
rivers of smoke in the region.



*Data availability*. CAMS data are freely available at the Copernicus Atmospheric Data Store
(https://ads.atmosphere.copernicus.eu/). AERONET station data are made freely available by the
NASA Goddard Space Flight Center (https://aeronet.gsfc.nasa.gov/). The SAM daily index is made
freely available by the NOAA Climate Prediction Center (https://www.cpc.ncep.noaa.gov/).
*Supplement.* The supplement related to this article is available online at XXXXXXXXXXX.
*Author contributions.* MG conceived the study, designed and performed the analysis and wrote the
paper. BP and MCAC performed the WR classification. All the authors contributed to the discussion
and interpretation the results and the writing of the text. PF and CF designed the original AEROCLO-
sA observational concept, and co-led the 5-year investigation.
*Competing interests*. PF is guest editor for the ACP Special Issue "New observations and related
modelling studies of the aerosol–cloud–climate system in the Southeast Atlantic and southern Africa
regions". The remaining authors declare that they have no conflicts of interests.
*Special issue statement.* This article is part of the special issue "New observations and related
modelling studies of the aerosol–cloud–climate system in the Southeast Atlantic and southern Africa
regions (ACP/AMT inter-journal SI)". It is not associated with a conference.
*Acknowledgments*. The authors thank the AERONET PIs (Brent Holben, Nichola Knox, Stuart Piketh,
Gillian Maggs-Kollin, Derek Griffith, and Willie Gunter) and their staff for establishing and
maintaining the AERONET sites used in this study, and K. Schepanski and F. Waquet for useful
discussion.
*Financial support*. The AEROCLO-sA project was supported by the French National Research Agency
under grant agreement n° ANR-15-CE01-0014-01, the French national program LEFE/INSU, the
Programme national de Télédetection Spatiale (PNTS, http://www.insu.cnrs.fr/pnts), grant n° PNTS-
2016-14, the French National Agency for Space Studies (CNES), and the South African National
Research Foundation (NRF) under grant UID 105958. The research leading to these results has
received funding from the European Union's 7th Framework Programme (FP7/2014-2018) under
EUFAR2 contract n°312609".



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



**Table 1.** AERONET station used in this study: locations and data availability (Version 3 Direct Sun algorithm, level2).

| Station | Country | Latitude | Longitude | Observations (coverage) |
|---|---|---|---|---|
| Ascension Island (AI) | UK Overseas Territory | 8.0°S | 14.4°W | 612 (2003-2017) |
| Bonanza (BO) | Namibia | 21.8°S | 19.6°E | 126 (2016-2017) |
| Gobabeb (GO) | Namibia | 23.6°S | 15.0°E | 219 (2015-2017) |
| Henties Bay (HB) | Namibia | 22.1°S | 14.3°E | 139 (2013-2017) |
| HESS (HE) | Namibia | 23.3°S | 16.5°E | 158 (2016-2017) |
| Simon's Town IMT (ST) | South Africa | 34.2°S | 18.4°E | 127 (2016-2017) |
| Upington (UP) | South Africa | 28.4°S | 21.2°E | 111 (2015-2016) |

**Table 2.** WR transition rate, computed as the fraction of transition from a WR (rows) into the others (columns). By definition, the diagonal represents persistence. Transition rates above 1/6 are reported in bold.

| WR | 1 | 2 | 3 | 4 | 5 | 6 |
|---|---|---|---|---|---|---|
| 1 | **0.39** | 0.08 | 0.08 | 0.03 | **0.28** | 0.14 |
| 2 | 0.06 | **0.61** | 0.06 | 0.08 | 0.11 | 0.06 |
| 3 | 0.04 | 0.12 | **0.45** | **0.24** | 0.06 | 0.09 |
| 4 | **0.29** | 0.12 | 0.03 | **0.40** | 0.03 | 0.13 |
| 5 | 0.03 | 0.12 | **0.31** | 0.02 | **0.46** | 0.05 |
| 6 | 0.05 | 0.10 | 0.08 | 0.07 | 0.10 | **0.59** |

**Table 3.** Circulation-to-environment characterisation: P-values of ANOVA and Kolmogorov-Smirnov test on AOD anomalies at the AERONET stations (Table 1). Values lower than 0.05 are reported in bold.

| Station | ANOVA | WR1 | WR2 | WR3 | WR4 | WR5 | WR6 |
|---|---|---|---|---|---|---|---|
| Ascension Island (AI) | **0.01** | 0.91 | 0.23 | 0.08 | 0.27 | 0.89 | **0.04** |
| Bonanza (BO) | 0.07 | 0.31 | 0.67 | 0.90 | **0.01** | 0.76 | 0.35 |
| Gobabeb (GO) | **<0.01** | **0.01** | 0.90 | 0.55 | 0.07 | 0.98 | **<0.01** |
| Henties Bay (HB) | **<0.01** | **<0.01** | 0.77 | 0.70 | 0.13 | 0.49 | 0.18 |
| HESS (HE) | **<0.01** | **<0.01** | 0.26 | **<0.01** | 0.93 | 0.71 | 0.36 |
| Simon's Town IMT (ST) | 0.09 | 0.26 | 0.27 | 0.91 | 0.87 | 0.13 | 0.61 |
| Upington (UP) | **<0.01** | **0.03** | 0.87 | **<0.01** | **0.02** | 0.64 | 0.23 |

**Table 4.** Environment-to-circulation characterisation: Chi-squared statistics for each quartile (Q1-4) at the AERONET stations (Table 1). Values exceeding 11.07, i.e. the critical threshold for the Chi-squared distribution with 5 degrees of freedom at 95% level of confidence, are reported in bold.

| Station | Q1 | Q2 | Q3 | Q4 |
|---|---|---|---|---|
| Ascension Island (AI) | 8.05 | 2.99 | **11.19** | 9.31 |
| Bonanza (BO) | 3.91 | 2.82 | 1.28 | 7.35 |
| Gobabeb (GO) | **12.29** | 1.99 | 10.68 | **17.03** |
| Henties Bay (HB) | 6.45 | 6.98 | 6.16 | **11.49** |
| HESS (HE) | **15.04** | 7.99 | **12.59** | **12.16** |
| Simon's Town IMT (ST) | 5.93 | 1.79 | 11.04 | **17.30** |
| Upington (UP) | **14.29** | 10.14 | 3.86 | **25.23** |

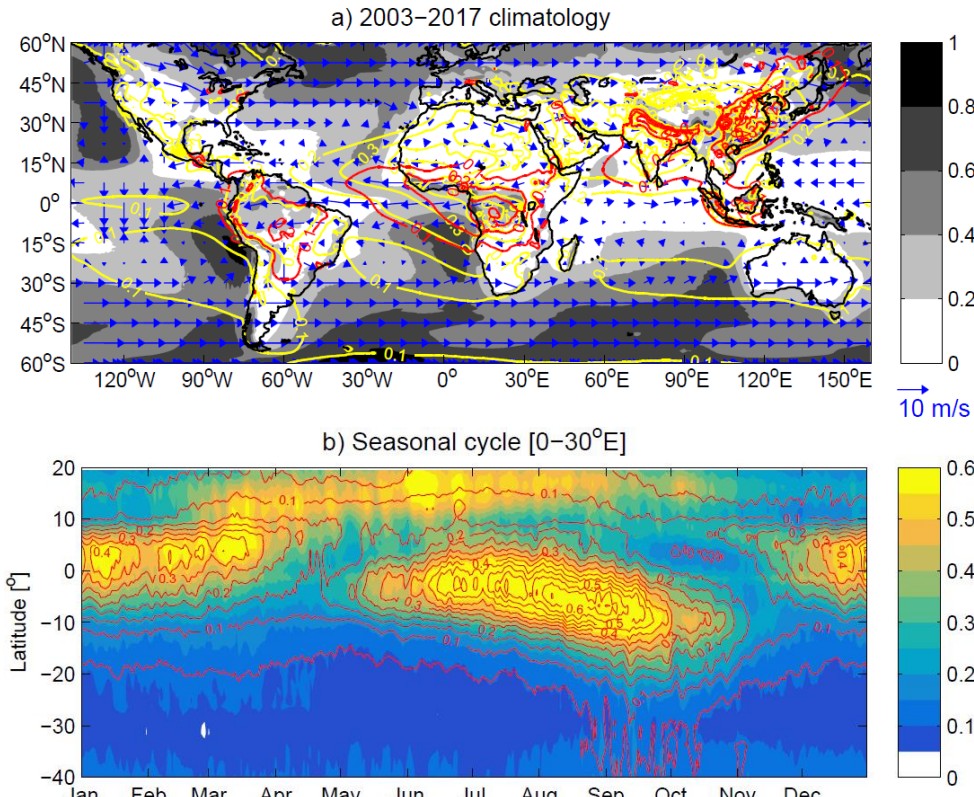

**Figure 1.** 2003-2017 climatology derived from CAMS reanalysis: (a) annual mean of total (yellow contours) and organic matter (red contours) AOD at 550 nm, low cloud cover fraction (shadings) and wind at 700 hPa (arrows); (b) annual cycle of total (shadings) and organic matter (red contours) AOD at 550 nm, averaged over [0-30°E].



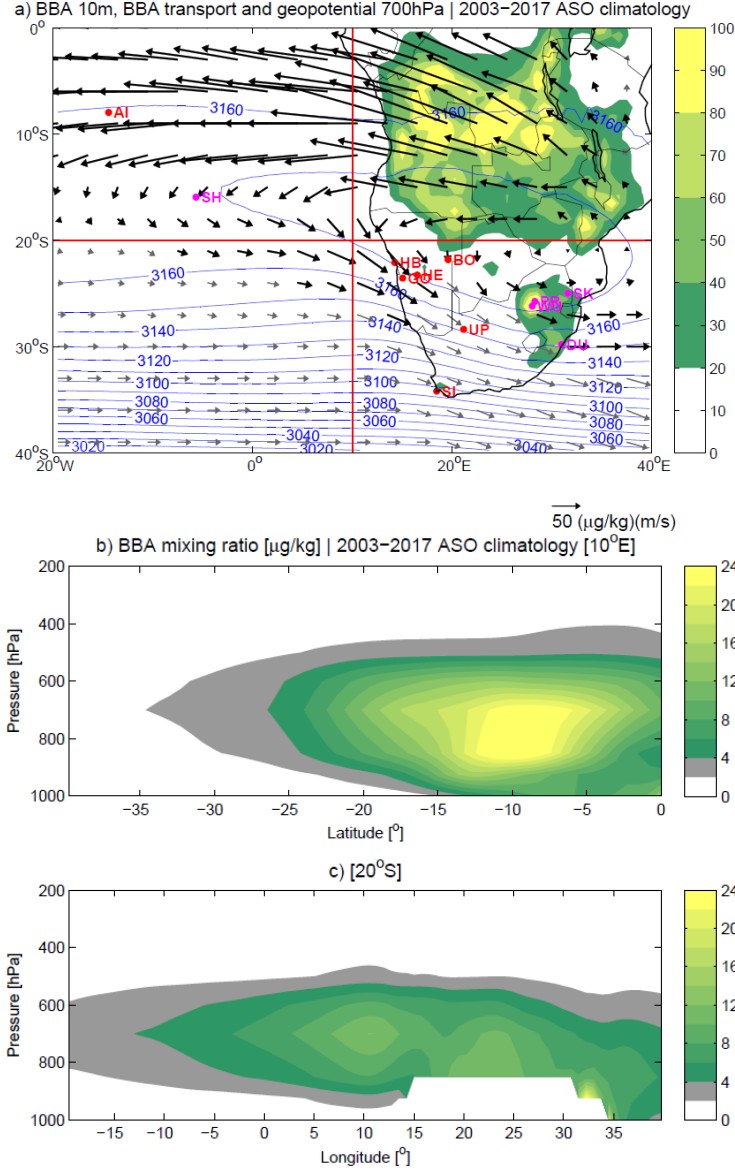

**Figure 2.** 2003-2017 ASO climatology derived from CAMS reanalysis: (a) BBA mixing ratio at 10m (µg/kg, shadings), BBA transport at 700 hPa (µg/kg m/s, arrows) and geopotential height at 700 hPa (m, contours); thick arrows highlight BBA transport corresponding to BBA mixing ratio greater than 4 µg/kg. Red lines indicate where BBA mixing ratio cross-sections are computed, red dots indicate the locations of the AERONET stations used in this study (see Table 1 for details), magenta dots indicate the locations of available stations not used in this study (see Section 2 for details). Vertical cross-sections of the BBA mixing ratio (µg/kg) at (b) 0°E and (c) 25°S.

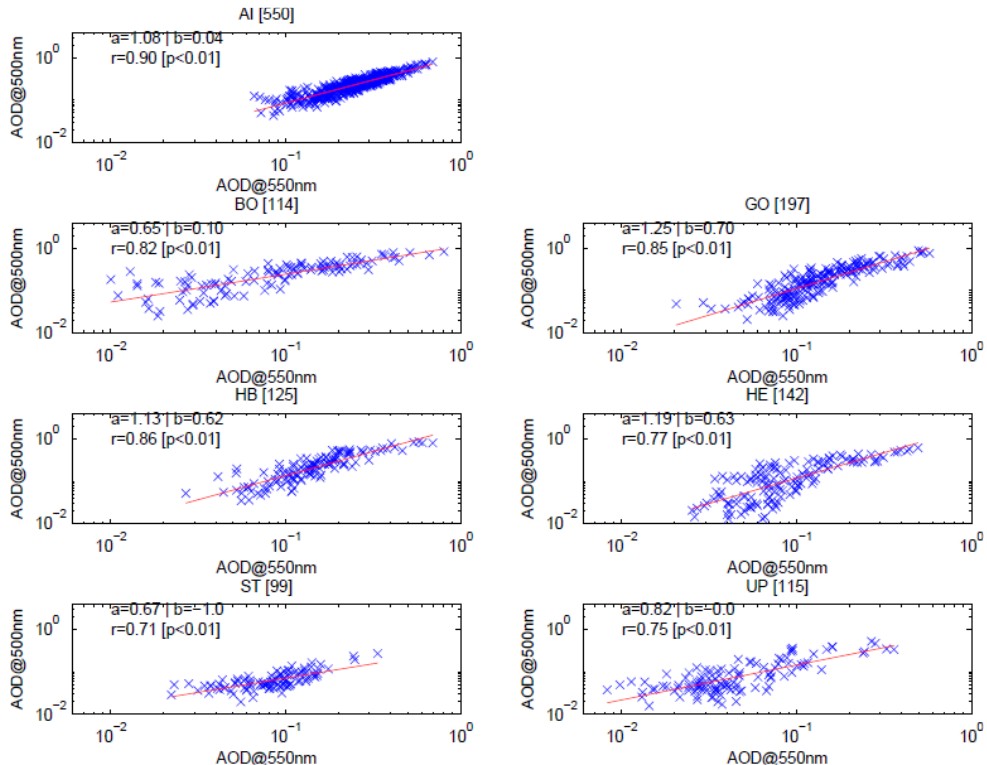

**Figure 3.** CAMS vs AERONET daily data comparison for ASO 2003-2017: CAMS reanalysis AOD at 550 nm vs AERONET observed AOD at 500 nm. CAMS data are extracted at grid points the closest to the station coordinates (Table 1). Red lines display the linear regression between CAMS and AERONET data, and the coefficients of the regression models are also reported in the plots, along with the correlation coefficient and the p-value. In titles, the size of the sample used in the linear regression model is reported in brackets (see Section 2 for details).

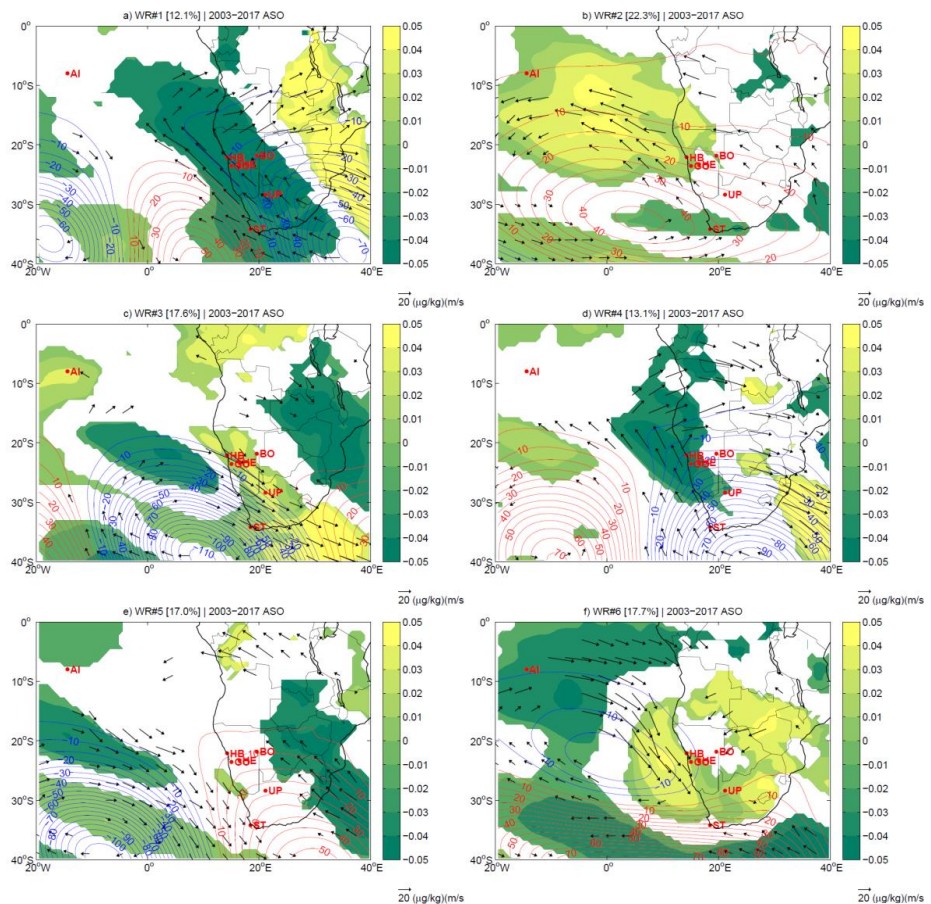

**Figure 4.** Anomaly patterns of CAMS geopotential height (m, contours), AOD at 550 nm (shadings) and BBA transport ((μg/kg)(m/s), arrows) at 700 hPa associated with the WRs classified from the geopotential height at 700 hPa in ASO 2003-2017. Frequency of the WRs is indicated in brackets; for AOD and BBA transport, only values significant at 95% level of confidence for a Student's t-test are displayed.



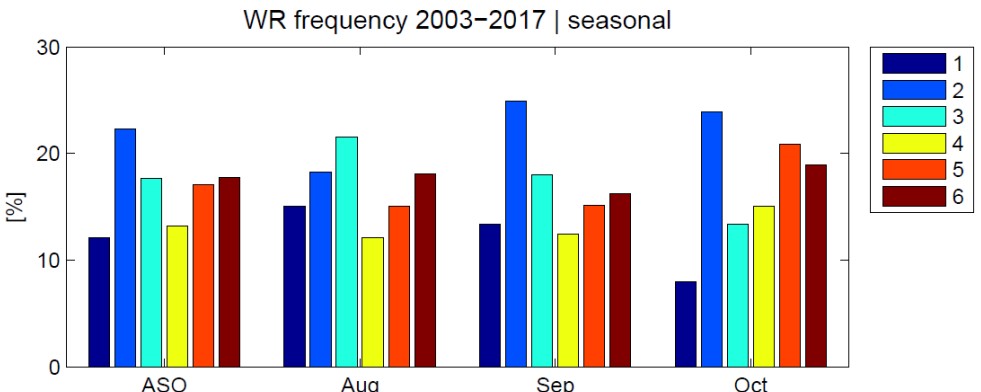

**Figure 5.** WR frequency 2003-2017: seasonal and monthly.



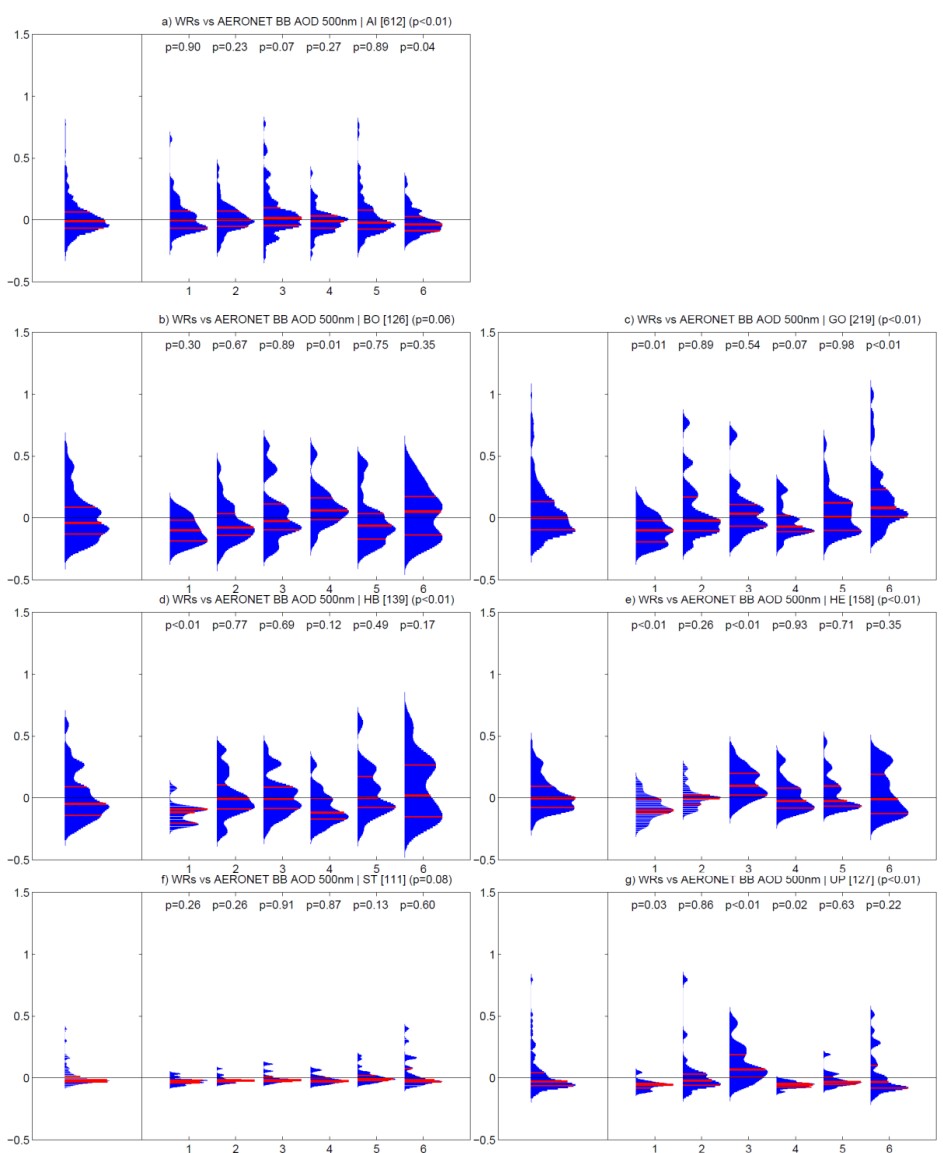

**Figure 6.** Circulation to environment characterisation: distributions of the AOD anomalies at 500 nm at the AERONET stations (Table 1), and as a function of the WRs. Probability density functions are estimated by using a normal kernel density; red lines represent 25th, 50th and 75th percentiles. For each WR, the p-value of a Kolmogorov-Smirnov test used to assess the difference with the total sample is reported. In titles, in brackets the number of available daily observations and the p-value of the ANOVA used to assess the WR characterisation are reported.

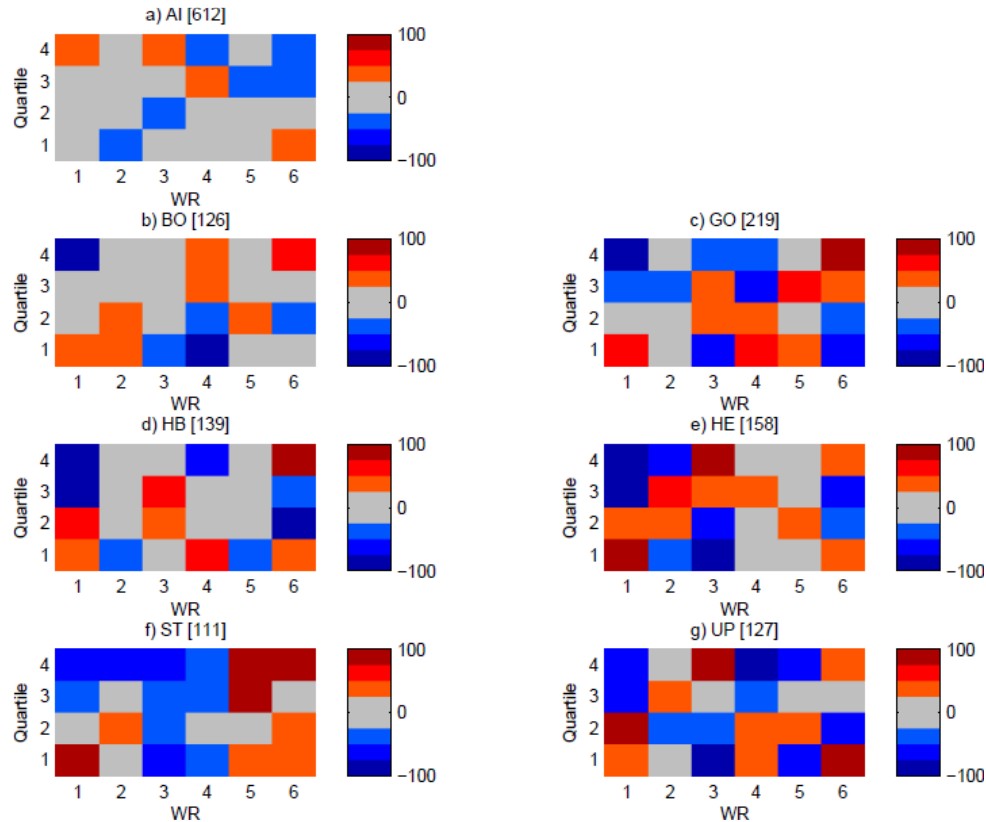

**Figure 7.** Environment to circulation characterisation: WR frequency anomaly as a function of the quartiles of the AOD anomalies at 500 nm at the AERONET stations (Table 1). Values are percentage changes relative to climatological frequencies. In brackets, the number of available daily observations are indicated.



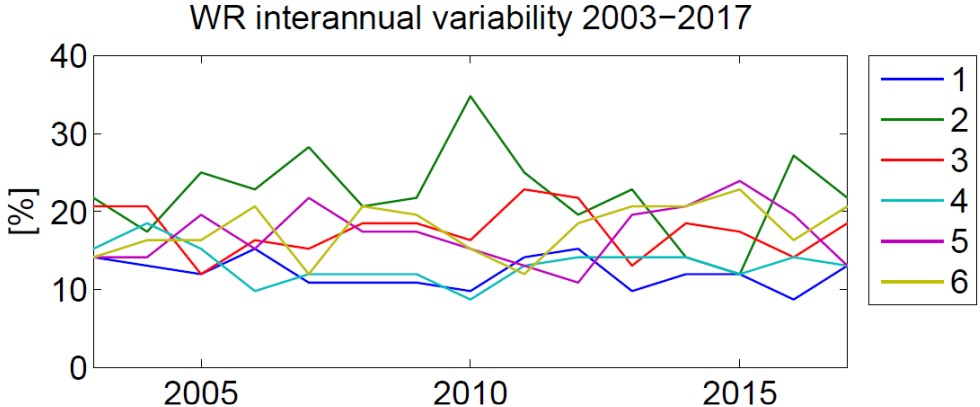

**Figure 8.** WR frequency 2003-2017: interannual variability.

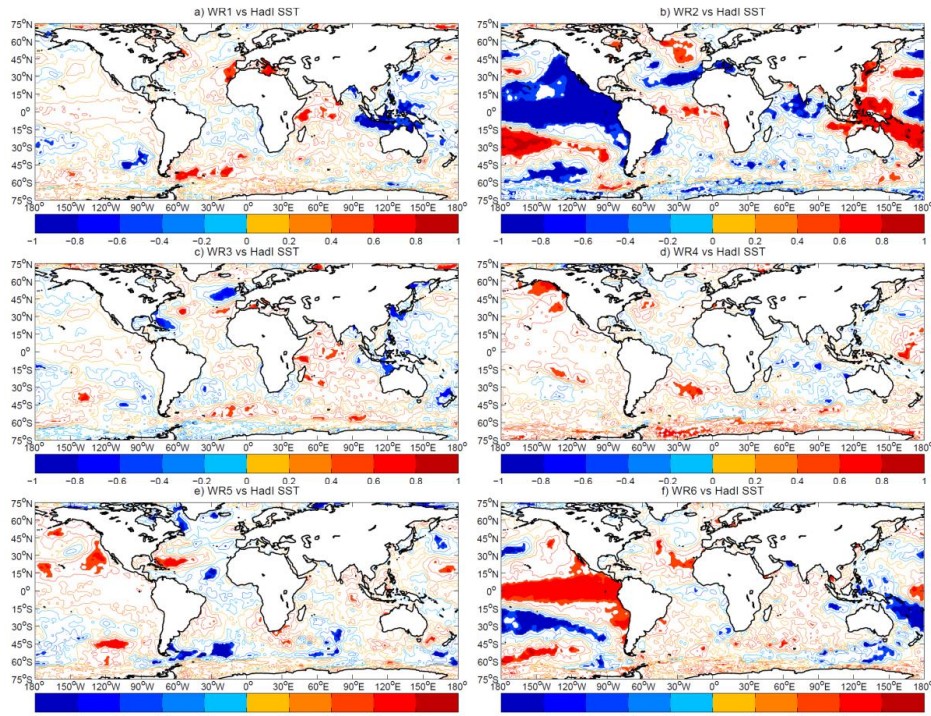

**Figure 9.** Interannual correlation over the period 2003-2017: WR frequency vs HadI SST in ASO.
Shadings display significant correlations at 95% level of confidence. Time series are detrended and
standardised.

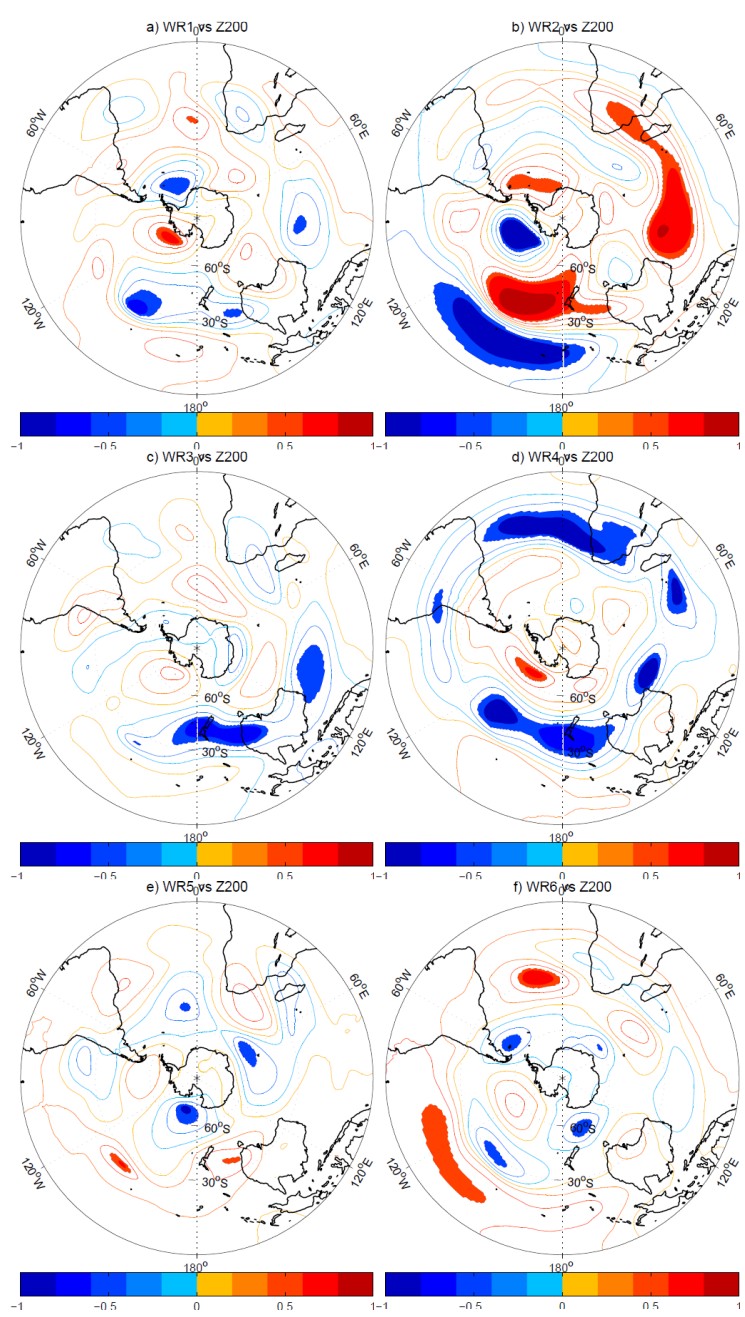

**Figure 10.** Interannual correlation over the period 2003-2017: WR frequency vs geopotential height at 200 hPa in ASO. Shadings display significant correlations at 95% level of confidence. Time series are detrended and standardised.



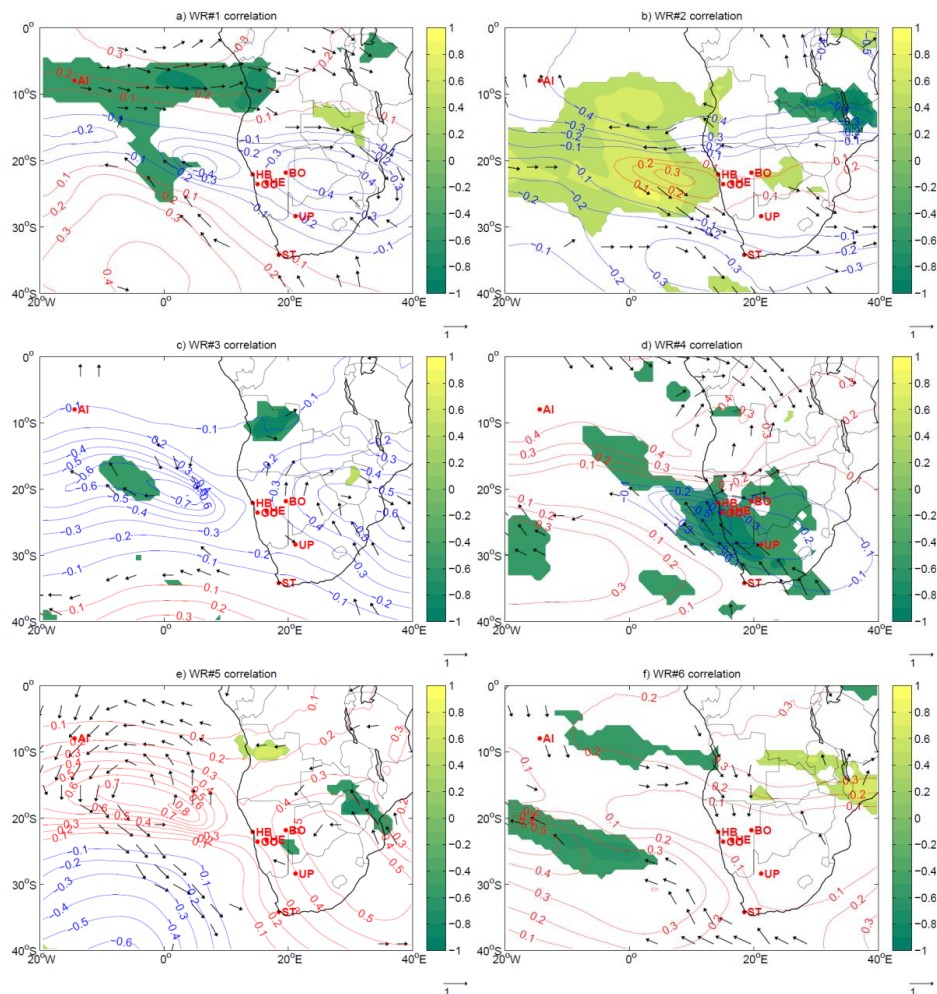

**Figure 11.** Correlation maps of WR frequency vs CAMS geopotential height (contours), AOD at 550 nm (shadings) and BBA transport (arrows) at 700 hPa. For AOD and BBA transport, only correlation significant at 95% level of confidence are displayed. Time series are detrended and standardised.