# Peer review of "A weather regime characterisation of winter biomass aerosol transport from southern Africa 2 3 Marco Gaetani (1,2); Benjamin Pohl (3); Maria del Carmen Alvarez Castro (4,5); Cyrille Flamant (6); 4 Paola Formenti (1) 5 6 (1) Université de Paris and Université Paris Est Creteil, CNRS, LISA, F-75013"

_Atmospheric Chemistry and Physics, 2021_

## Author Comment (AC1)

**General Comments**

The authors present a classification of the circulation over southern Africa and the South Atlantic into six weather regimes (WRs). They link these WRs with the transport of biomass burning aerosol (BBA) via the corresponding wind field and with the aerosol optical depth (AOD). Furthermore, they investigate whether there are significant links between the observed AOD at surface stations and the WRs.

This study appears to be the first linking WRs to aerosol transport in this region. It provides valuable insight in the circulation over the South Atlantic and southern Africa and shows significant relations between AOD at surface stations and some of the WRs. When WR classifications have been performed in the Northern Hemisphere, e.g. the Euro-Atlantic sector, these regimes are quite persistent and describe variability on timescales of 5-15 days, i.e. for a large part beyond synoptic timescales, while most WRs studied here describe precisely that synoptic variability in the region. Therefore, the regimes in this region are more a useful descriptive tool for the circulation than persistent features in phase space. Nevertheless, these WRs prove to be well suited for studying BBA transport in this region and could have a lot of potential for studying other circulation processes affecting southern Africa. The authors get this across quite well, although I think the potential of the WR can be emphasized more.

We thank the Reviewer for her/is interest in our work and the time s/he spent in making constructive comments, which helped us in substantially improving the quality of the manuscript. We provide below a point-by-point response to all the comments/requests. Reviewer's comments are in black, our responses are in blue.

**Specific Comments**

Can you comment on the robustness of the WRs? The time series used is relatively short, being only 15 years of ASO data, for the classification of WRs. It would be good to know the WRs are robust with respect to the reanalysis products used by computing the WRs for e.g. ERA-5 which has a longer available time period.

Are the WRs you obtain dependent on the selected domain? Most of the variability, especially in WR1, 3, 4 and 5, is south of 20S. Do the WRs change if you e.g. only take the domain up to 10S, or extend the domain beyond 40S? A similar question on the number of PCs, have you looked into any dependence of the WRs on this?

We agree with the reviewer that an assessment of the robustness of the WR classification is needed. However we believe that the synoptic characterisation of the aerosol transport should be performed by using the CAMS product, which assure coherence between atmospheric circulation and aerosol data. Therefore we decide to keep using the CAMS classification to characterise the aerosol transport, and perform an additional assessment of its robustness following your suggestions. We tested: the sensitivity of the WR classification in CAMS to changes in the geographical domain and the number of retained PCs. Moreover the WR classification defined in 2003-2017 is compared with a WR classification defined in 1981-2020 by using ERA5 data. Section S2 has been added in the Supplement to discuss the sensitivity of the CAMS classification to changes in the geographical domain and the number of retained PCs, WR centroids and frequencies from the ERA5 classification have been added to Fig. 4 and 7, and a long paragraph has been added to Section 3.1 to discuss the classification in ERA5 and summarise the sensitivity tests. We can conclude that:

*"The sensitivity tests performed on the WR classification of the CAMS data show a high degree of robustness with respect to changes in the time period, and a good degree of robustness with respect to changes in the geographical domain and the retained variance, highlighting that the classification well represents the main features of the synoptic circulation in the region".*

You comment on the limited coverage of the CAMS reanalysis not being sufficient for defining a climatological seasonal cycle (p4, 100-102). As you only consider three months of data, would it make sense to simply use a fixed climatology, i.e. average over all months, with respect to which to compute the anomalies? And would this affect the WRs?

Removing the ASO climatology would be problematic because the emission of BBA from the source region (see the organic matter mixing ratio at 10m, averaged in the emission region in Tropical Africa, Figure R1), and consequent transport (see Fig. 1) largely vary from August to October, in terms of both intensity and variability. Therefore, the removal of a fixed climatology would result in biased daily anomalies, too high in August and too low in October. We believe that the definition of daily anomalies as the high frequency component of the day-by-day variability is well suited for studying the synoptic variability of both circulation and BBA transport in ASO.

[Figure]

**Figure R1.** Climatology of the organic matter mixing ratio at 10 m in the emission region in Tropical Africa, used to estimate the intensity of BBA emission.

In your study you use 6 WRs, and in the supplement you also show results for 7 WRs. Why did you choose to work with 6 WRs in the main manuscript instead of 7? If it's because the results simply are better, could you state that more clearly in the main manuscript. In the end that's the aim of using the WRs. From my point of view, selecting 6 only because it's the lowest significant number of WRs would already be a valid argument. Also, I'm quite surprised 2 WR yields such a poor classificability index (supplement), can you comment on this?

We first highlight that the red-noise test shows that the 6 and 7 class partitions are both significant and correspond to physically coherent atmospheric patterns, and partitioning the synoptic variability into 6 or 7 regimes would be equally appropriate. The choice of focusing on the 6 WR classification is mainly based on the fact that this is easy and convenient to interpret: it results in one high-pressure regime, one low-pressure regime (denoting contrasted intensities in the strength of the South

Atlantic/St. Helena High), and 4 regimes depicting transient disturbances in the middle latitudes, i.e. the propagation and life cycle of temperate waves embedded in midlatitude dynamics, and materializing the synoptic-scale variability that develops there. In addition, the comparison between the 6 and 7 class partitions shows that the 6 WR classification performs (slightly) better in characterising the BBA transport. This is now clearly stated at the end of Section 2.3:

*"The synoptic characterisation of the BBA transport is performed by using both the 6-class and the 7-class partitions. This study focuses on the 6-class partition, i.e. the classification with the lowest significant number of WRs, which leads to physically coherent atmospheric patterns describing the main features of the synoptic variability (see Section 3.1). Furthermore, the comparison between the 6 and 7 class partitions shows that the 6 WR classification performs better in characterising the BBA transport in the region (see Section S3)".*

Concerning the low classificability index associated with the 2-class partitioning, we do agree that in most cases it is much larger than here (e.g., Pohl et al. 2021, Supplementary Material, https://journals.ametsoc.org/view/journals/apme/aop/JAMC-D-20-0255.1/JAMC-D-20-0255.1.xml). A possible reason for this low score is that two regimes could not perform well to depict transient wave activity in the southern part of our domain. Yet, such transient disturbances are, by far, associated with the largest variance of the input field (geopotential height anomalies), attracting 4 regimes out of 6 to depict the eastward propagations of such synoptic disturbances. Consequently, these regimes are statistically instable and weakly coherent, because they are unable to depict well the dominant modes of variability in this region.

Could you expand on the computation of BBA emission and transport? For the transport, are you discussing anomalies with respect to the low frequency component (season-wise) or absolute values? Similarly, for AOD, are you plotting anomalies? It would be good to clarify throughout.

As stated in Section 2.1:

*"The BBA emission is estimated by the organic matter mixing ratio at 10m, the BBA transport is estimated as the product of organic matter mixing ratio and wind at 700 hPa, and the aerosol spatial distribution is represented by the aerosol optical depth (AOD) at 550 nm (Fig. 2a)".*

All the data from ERA5, CAMS and AERONET are used in the synoptic classification and characterisation (Fig. 4, 6, 9, 10 and 11) as daily anomalies, computed by removing the low frequency component, as stated in Section 2.1 and 2.2. The nature of the displayed variables is also stated in figure captions. In the revised version of the manuscript, in Section 2.3 we further specify that daily anomalies are used in the PCA:

*"The atmospheric circulation is first characterised by isolating the main modes of variability represented by the empirical orthogonal functions (EOFs) derived from a principal component analysis (PCA) of the **geopotential height daily anomalies**".*

In Section 2.4, we state that:

*"1) **Daily AOD anomalies** are linked to the corresponding WR and grouped, and statistical differences among groups are investigated (circulation-to-environment approach, C2E)".*

*"2) **Daily AOD anomalies** are divided into quartiles, and the changes in the WR occurrences within each quartile are studied (environment-to-circulation approach, E2C)".*

The WRs come in two types, with 2 and 6 describing the oscillation of the South Atlantic pressure field and the other four representing propagating disturbances along the midlatitude mean flow. The

latter four are not very persistent, and represent a travelling wavetrain (1-5-3-4). How do these two sets of WRs relate and could you also study the BBA transport using a travelling wave perspective?

Thank you for this comment, which prompted us to do some additional analysis adding robustness to our findings. Persistence is studied by analysing the length of the WR sequences, showing short persistence for both the SA and ML regimes (Fig. 5), and indicating synoptic variability as the dominant feature of the atmospheric variability in the region. A composite analysis from the occurrence of the WR (day 0) to day +4 shows that the SA regimes are characterised by pressure oscillations, with around 3 day lifetime, while the ML regimes represent a propagative pattern, with around 4 day lifetime (ML1 → ML2 → ML3 → ML4, see Fig. 6). This is now discussed in Section 3.1. The synoptic variability associated with the WR occurrence is also reflected in the BBA transport, as illustrated by a lead-lag correlation analysis of the CAMS AOD in the South Atlantic and southern Africa (Fig 9). The lead-lag correlation shows *"8 day period for the AOD anomalies to build up in the tropical South Atlantic along the easterly route from tropical Africa"*, and *"a 6 day period for the river of smoke to build up in the South Atlantic and move eastward across southern Africa"*, as discussed in Section 3.2.

You look into the sub-seasonal variability of the WR regime frequencies (sec 3.1, p7). I am unsure whether you have sufficient data to state that these differences are robust, as noise will affect these results. Specifically, if you have insufficient data to define a seasonal cycle for it is affected by interannual variability, do you have enough to do something similar for the WR frequencies? I.e. on average you have 15x30/6 = 75 days in each WR during one month, to which changes of 5-10 days in occurrence can have quite an impact.

We do agree with the Reviewer that there are not enough data in CAMS for assessing the intraseasonal variability of WR frequencies. In the new version of the paper, the seasonal frequency of CAMS WRs is presented in Fig. 7. In addition, we show in Fig. 7 the seasonal and intraseasonal frequencies of the ERA5 WRs, computed across 40 years, from 1981 to 2020:

*"The availability of 40 year time series in ERA5 allows to robustly estimate WR frequencies at the intraseasonal time scale (Fig. 7). Differences are limited to 1-2%, with the exception of ML2 and ML3, increasing by 3% and decreasing by 4% during the season, respectively".*

Can you expand the discussion of the link between your WR and the SAM, which you briefly mention (top of p8)? Why do you think WR6 relates better to the SAM than WR2? As WR2 overall occurs more often, it would be interesting to know any discrepancies. Can you include the results that are not shown in the supplement?

The analysis of the WR-SAM relationship has been revisited by using the long term ERA5 classification, and so the discussion of the results. The WR-SAM relationship has been analysed on daily and interannual time scales by comparing with the SAM daily and monthly indices provided by the NOAA Climate Prediction Centre (https://www.cpc.ncep.noaa.gov/products/precip/CWlink/daily_ao_index/aao/aao.shtml). The daily connection has been explored by means of the C2E and E2C approaches and a new Figure 8 has been added to the main text. The analysis of the daily variability in the period 1981-2020 now shows that more persistent WRs (SA+ and SA-) are associated with positive SAM phases, i.e. with weaker westerlies at midlatitudes. Conversely, the "travelling" WRs (ML1-4) are associated with negative SAM phases, i.e. with stronger westerlies. The discussion in Section 3.1 has been expanded.

The WR-SAM interannual relationship has been also investigated by correlating the 1981-2020 WR frequency with the SAM index and the 700 hPa geopotential height. No significant signal is detected

(see Fig. R2), as expected because of the synoptic variability dominating the WRs. A sentence has been added to Section 3.4:

*"The analysis of the WR-SAM relationship at the interannual time scale shows poor results when the WR frequency time series are correlated with the SAM monthly index. Similarly, the correlation between the WR frequency time series and the monthly averages of the geopotential height at 700 hPa in the Southern Hemisphere does not show evident correlation patterns (not shown)".*

[Figure]

**Figure R2.** ERA5 WR frequency vs SAM index, for ASO 1981-2020.

Would there be any effect of the duration of the WRs onto the CAMS AOD, and that of the surface stations? I can imagine that if a regime lasts longer the effect on AOD is stronger as well, more so for WR2 and 6. Also, is it possible that there is an effect of past WRs (i.e. delayed) onto the measured AOD, possibly linking to the discrepancies between surface station AOD values and those of CAMS?

The analysis of WR sequences shows short persistence, highlighting the dominance of the synoptic variability in the atmospheric and aerosol dynamics, and suggesting limited impact of persistence on AOD anomalies at surface stations. The transient pattern characterising the ML WRs suggests the possibility of predicting AOD anomalies with a 2-3 day lead time. However, the C2E characterisation of in situ observations shows little improvement for 1 and 2 day lead times (see Fig. R3 and R4), and ambiguous results for 4 day lead time (see Fig. R5). We agree that the predictive performance of the WR classification is worth of being further investigated, however the limited availability of in situ measurements prevents a robust skill assessment. Such a research could be done on data with larger time-space coverage (e.g. satellite products), but it is beyond the scope of this paper.

[Figure]

**Figure R3.** Circulation to environment characterisation: for each AERONET station (Ascension Island, Bonanza, Namibian Stations, Upington), (left panel) distribution of the AOD anomalies at 500 nm, and (right panel) for each CAMS WR, with 1 day lead time. Anomaly distributions significantly different from the climatological sample are displayed in green (p-values of the Kolmogorov-Smirnov test used to assess the significance of the differences are reported in Table S5). In titles, the number of available daily observations and the p-value of the ANOVA used to assess the WR characterisation are reported in brackets.

[Figure]

**Figure R4.** Same as Fig. R3, but for 2 day lead time.

[Figure]

**Figure R5.** Same as Fig. R3, but for 4 day lead time.

Three (maybe four) stations are located close together (GO, HB, HE, and BO(?)). Differences in AOD distributions (Fig 6) for each of the WR thus could be primarily caused by local effects, e.g. for WR3, can you comment on this? Similarly for the WR frequency anomalies in Fig 7. Also, would it make sense to pool these stations together to get more robust statistical results (I am not familiar with the data itself, so do not know whether this would be sensible)?

Thank you for the suggestion. Indeed, the three stations in western Namibia are characterised by slightly different local conditions, which could add local effects to the AOD measurements. Namely, Henties Bay is a coastal site, exposed to both marine and mineral dust aerosols; Gobabeb is in the Namib Desert, exposed to mineral dust aerosols; HESS is located inland in the savannah, exposed to possible local sources of BBA. Therefore, to filter out possible local effects, we built a time series associated with the three stations (referred as Namibian Stations, NS) computed as the average AOD when observations are available in at least 2 stations out of 3. This also allows to have a time series longer than the three individuals (276 observations, spanning from 2013 to 2017). The combined time series is presented in Section 2.2, the analysis is added to Fig. 3, 10 and 11, and results are discussed in the main text as representative of mean conditions in western Namibia (Section 3.3).

More generally, station data is limited. Can you comment on how many days are in each of the WR for the different stations? This would help the reader get an idea of the robustness of the results in Fig 6 and 7.

The number of available observations for each station is reported in Table 1 and the number of data points used in the analysis is displayed in Fig. 3, 10 and 11. In Section 2.2 we highlight that stations with less than 100 observations are discarded:

*"Among the available stations, St. Helena (15.9°S, 5.7°W) and Wits University (26.2°S, 28.0°E) are not included because of the limited coverage (less than 100 observations during the study period)"*.

In the revised version, a sentence has been added at the beginning of Section 3.3 to highlight this aspect:

*"It is highlighted that data availability and coverage in most of the stations is limited (see Table 1), resulting in circa 20-40 observations per WR on average. Only the station in Ascension Island covers the whole period analysed, providing more than 600 observations, i.e. circa 100 observations per WR on average"*.

In the Conclusions we also state that:

*"Limited data availability in most of the stations prevents a robust statistical validation of the synoptic characterisation of observations at the regional scale"*.

How robust are your results on interannual variability, having only ~90 days with associated regimes for each year? In my experience interannual variability in regime frequencies of reanalysis products can be very large, and does not necessarily allow for clearly showing a signal through the noise. Could you use a longer reanalysis to look into links with e.g. ENSO to make these links more robust? I am not surprised WR 1,3,4,5 do not show any significant links with SST, as these are short-lived regimes. Have you thought about the persistence effect of the regimes and whether that could be affected (I appreciate there's too little data for this)?

Thank you for this comment. After building the WR classification in ERA5, which results being consistent with the analysis of CAMS data, in the revised version of the manuscript we use this longer time series for analysing the interannual variability across 40 years. New results show no large

differences among the WRs, with no significant trends and comparable variability (STD between 3% ad 4%). We highlight that the WR classification shows dominant synoptic variability and short persistence, therefore we expect no strong signal at the interannual time scales (e.g. there is no relationship with the interannual SAM). When the WR-SST teleconnection is explored, a weak relationship is found between SA regimes (showing some persistence) and SST anomalies in the tropical Pacific, along with a possible teleconnection pattern (see Fig. 13 and Section S4). The analysis of the WR-AOD interannual relationship has been suppressed because of the limited AOD data coverage, in both the reanalysis products and in situ observations.

Most plots are not very clear and take a long time looking at to understand what they show. Could you add labels to all axes and colorbars where they are not there, increase the fontsize of ticks, labels and captions, and remove any redundant information. The z700 patterns of the regimes are not clear at the moment (Fig 4), maybe increasing the line thickness would help, or otherwise I think it would be good to add them in a separate figure, as they are important to get across well. For Fig 6 and 7 it would be good to clearly see which results are significant and which are not, e.g. change the colour in Fig 6 and lines around the relevant cells in Fig 7. Moreover, there is some discrepancy between the p-values given in Fig 6 and Tab 3. I think it is sufficient to show these values only in the table if you indicate the significant ones in another way.

The quality of the figures has been improved. Circulation patterns are shown by thicker contours and WRs are also shown in separate figures without superimposed BBA (see Fig. S4 and S7). Fig. 10 and 11 have been modified, highlighting the significant differences. Tables with ANOVA, Kolmogorov-Smirnov and chi-squared statistics have been moved to Section S5 in the Supplement.

**Technical Corrections**

Could you rearrange the WRs such that their order links better to the regimes themselves? For example, change WR2, 6 to WR1 and 2, and WR1, 5, 3, 4 to WR3, 4, 5, 6 (in order of transitions), respectively. It would also be good to line up the 7 WRs in the supplement with the six in the manuscript.

Thank you for this useful suggestion. The WRs have been rearranged and labelled accordingly to their circulation features: SA+ and SA- for the pressure anomalies in the South Atlantic, ML1-4 for the midlatitude pressure anomalies. The ML regimes are labelled to highlight the propagative nature of the transition, with pressure anomalies in ML1 moving eastward into ML2, 3, and 4. The 7 class WR classification has been labelled following the same principle, and the WRs from both the classifications are now displayed in the same figure in the Supplement (Figure S7 in Section S3).

Please check the figure numbers as they are not everywhere correct, e.g. p9, 268, Fig. 6a/6b should be Fig 7a/7b and p11, 318, Fig. 10 should be Fig. 11. I might have missed some others.

Figures in the main text have been reorganised, and new figures have been added. The text-figure correspondence has been checked and fixed where necessary.

Throughout the manuscript: South Atlantic -> the South Atlantic

This has been corrected where necessary, thank you.

p2, 27 by -> of

Corrected.

p2, 35-39 Partly repeating what is mentioned earlier in the abstract, I would recommend rewriting or weaving it into the earlier part of the abstract

The last paragraph of the abstract has been rephrased:

*"The skill in characterising the BBA transport shown by the WR classification indicates the potential for using it as a diagnostic/predictive tool for the aerosol dynamics, which is a key component for the full understanding and modelling of the complex radiation-aerosol-cloud interactions controlling the atmospheric radiative budget in the region."*

p3, 44 Remove "on"

Removed.

p3, 47 originated -> originating

Changed.

p3, 48 Insert "and is" between "(Fig. 1a)" and "a prominent"

Done.

p3, 52 definition -> term

Changed.

p3, 59 shed light -> shed light on

Corrected.

p3, 63 the Antarctica -> Antarctica

Corrected.

p4, 76-78 This sentence is nearly the same as the one above

Yes, the sentence has been removed.

p5, 106 You already mention AOD here as an abbreviation but only define it in line 114, also it's already mentioned in the caption of Fig 1, so might be good to clarify it there as well

The acronym is now defined here and in the caption of Fig. 1.

p5, 124 with -> of

Corrected.

p5, 131 as linear -> to be linear

Changed.

p6, 139-140 before to perform the linear regression -> before the linear regression is performed

Changed.

p6, 148 to mask -> mask

Corrected.

p7, 182 southward the midlatitude westerly flow -> the midlatitude westerly flow southward

Changed.

p7, 182-187 Do you have any references supporting this, or is it solely based on Fig 2a?

This is basically the description on Fig. 2a, which represents the climatology for the ASO season. A reference to Adebiyi and Zuidema (2016), describing the atmospheric circulation and aerosol climatology in September-October, has been added.

p7, 189 filed -> field

Corrected.

p7, 192 the WR2 -> WR2

The sentence has been rephrased.

p7, 203 Can you change the order of the preferred transitions in the brackets, so they are in order of transitioning?

This sentence has been rephrased, and the WR transitions are organised to highlight the eastward propagative pattern: ML1 → ML2 → ML3 → ML4.

p8, 215 coherently -> coherent

The sentence has been removed.

p8, 216 results statistically -> results are statistically

The sentence has been removed.

p9, 249 Can you repeat the null hypothesis here?

Done.

p12, 357 WR then clustering -> WR clustering

Corrected.

**Table 1.** AERONET station used in this study: locations and data availability (Version 3 Direct Sun algorithm, level2).

| Station | Country | Latitude | Longitude | Observations (coverage) |
|---|---|---|---|---|
| Ascension Island (AI) | UK Overseas Territory | 8.0°S | 14.4°W | 612 (2003-2017) |
| Bonanza (BO) | Namibia | 21.8°S | 19.6°E | 126 (2016-2017) |
| Namibian Stations (NS) | Namibia | | | 276 (2013-2017) |
| Gobabeb (GO) | Namibia | 23.6°S | 15.0°E | 219 (2015-2017) |
| Henties Bay (HB) | Namibia | 22.1°S | 14.3°E | 139 (2013-2017) |
| HESS (HE) | Namibia | 23.3°S | 16.5°E | 158 (2016-2017) |
| Simon's Town IMT (ST) | South Africa | 34.2°S | 18.4°E | 127 (2016-2017) |
| Upington (UP) | South Africa | 28.4°S | 21.2°E | 111 (2015-2016) |

**Table 2.** WR transition rates in the CAMS classification, computed as the percentage of transitions from a WR (rows) towards the others (columns). By definition, the diagonal represents persistence. Transition rates above 1/6, i.e. the threshold for non-random transitions, are reported in bold.

| WR | SA+ | SA- | ML1 | ML2 | ML3 | ML4 |
|---|---|---|---|---|---|---|
| SA+ | **61** | 6 | 6 | 11 | 6 | 8 |
| SA- | 10 | **59** | 5 | 10 | 8 | 7 |
| ML1 | 8 | 14 | **39** | **28** | 8 | 3 |
| ML2 | 12 | 5 | 3 | **46** | **31** | 2 |
| ML3 | 12 | 9 | 4 | 6 | **45** | **24** |
| ML4 | 12 | 13 | **29** | 3 | 3 | **40** |

**Table 3.** WR transition rates in the ERA5 classification, computed as the percentage of transitions from a WR (rows) towards the others (columns). By definition, the diagonal represents persistence. Transition rates above 1/6, i.e. the threshold for non-random transitions, are reported in bold.

| WR | SA+ | SA- | ML1 | ML2 | ML3 | ML4 |
|---|---|---|---|---|---|---|
| SA+ | **58** | 8 | 4 | 13 | 6 | 10 |
| SA- | 7 | **60** | 9 | 10 | 8 | 6 |
| ML1 | 7 | 18 | **42** | **21** | 8 | 4 |
| ML2 | 11 | 5 | 3 | **50** | **27** | 3 |
| ML3 | 4 | 12 | 6 | 4 | **45** | **28** |
| ML4 | 14 | 7 | **23** | 5 | 5 | **46** |

[Figure]

**Figure 1.** 2003-2017 climatology derived from CAMS reanalysis: (a) annual mean of total (yellow contours) and organic matter (red contours) aerosol optical depth (AOD) at 550 nm, low cloud cover fraction (shadings) and wind at 700 hPa (arrows); (b) annual cycle of total (shadings) and organic matter (red contours) AOD at 550 nm, averaged over Africa and South Atlantic [0-30°E].

[Figure]

**Figure 2.** 2003-2017 ASO climatology derived from CAMS reanalysis: (a) organic matter mixing ratio at 10m (μg/kg, shadings), organic matter transport at 700 hPa (μg/kg m/s, arrows) and geopotential height at 700 hPa (m, contours); vertical cross-sections of the organic matter mixing ratio (μg/kg) at (b) 0°E and (c) 25°S. In (a), thick arrows highlight organic matter transport corresponding to organic matter mixing ratio greater than 4 μg/kg; red lines indicate where organic matter mixing ratio cross-sections are computed; red dots indicate the locations of the AERONET stations used in this study (see Table 1 for details); magenta dots indicate the locations of available stations not used in this study (see Section 2.2 for details).

[Figure]

**Figure 3.** Daily data comparison for ASO 2003-2017: CAMS reanalysis AOD at 550 nm vs AERONET observed AOD at 500 nm. CAMS data are extracted at grid points the closest to the station coordinates (see Table 1). Red lines display the linear regression between CAMS and AERONET data, and the coefficients of the regression models are also reported in the plots, along with the correlation coefficient and the p-value. In titles, the size of the sample used in the linear regression model is reported in brackets (see Section 2.2 for details).

[Figure]

**Figure 4.** Left panels: anomaly patterns of CAMS geopotential height (m, contours), AOD at 550 nm (shadings) and organic matter transport ((μg/kg)(m/s), arrows) at 700 hPa associated with the WRs classified from CAMS geopotential height at 700 hPa in ASO 2003-2017. Right panels: anomaly patterns of ERA5 geopotential height (m, contours) at 700 hPa associated with the WRs classified from ERA5 geopotential height at 700 hPa in ASO 1981-2020. Dots indicate the locations of the AERONET stations used in this study. Frequency of the WRs is indicated in brackets. For AOD and organic matter transport, only values significant at 95% level of confidence after a Student's t-test are displayed.

[Figure]

**Figure 4.** Continued.

[Figure]

**Figure 5.** WR persistence in (a) CAMS and (b) ERA5 classifications, displayed as the distribution of the WR sequences. Red lines represent the median, boxes represent the interquartile range, whiskers extend up to the 1.5 of the interquartile range, outliers are displayed as red crosses.

[Figure]

**Figure 6.** Daily evolution of the 700 hPa geopotential height anomalies (m, contours) associated with the CAMS WR classification, computed as composites from the WR occurrence (day 0) to day +4.

[Figure]

**Figure 7.** WR frequency in CAMS (computed in the period 2003-2017) and ERA5 (computed in the period 1981-2020) classifications.

[Figure]

**Figure 8.** (a) Circulation to environment characterisation (C2E): (left panel) daily Southern Annular Model (SAM) index distribution, and (right panel) for each ERA5 WR. Probability density functions are estimated by using a normal kernel density; red lines represent 25th, 50th and 75th percentiles. Anomaly distributions significantly different from the climatological sample are displayed in green (significance is assessed by a Kolmogorov-Smirnov test at 95% level of confidence). The p-value of the ANOVA used to assess the C2E characterisation is reported in brackets. (b) Environment to circulation characterisation (E2C): ERA5 WR frequency anomaly for each quartile of the daily SAM index. Values represent percentage changes relative to climatological frequencies. Green boxes highlight significant changes in the WR frequencies, i.e. frequency anomalies exceeding the critical threshold (11.07) for the chi-squared statistics with 5 degrees of freedom at 95% level of confidence (see Section 2.4 for details).

[Figure]

**Figure 9.** Daily lead-lag correlation analysis of CAMS AOD anomalies. AOD anomalies averaged in the (a) South Atlantic and (c) southern Africa are correlated with themselves: dotted lines display individual year correlations; South Atlantic and southern Africa domains are displayed as boxes in (b) and (d), respectively. Correlation maps: AOD anomalies averaged in the (b) South Atlantic and (d) southern Africa are correlated with the AOD anomalies in the South Atlantic/southern Africa domain. Correlations are computed year-by-year and averaged.

[Figure]

**Figure 10.** Circulation to environment characterisation: for each AERONET station, (left panel) distribution of the AOD anomalies at 500 nm, and (right panel) for each CAMS WR. Probability density functions are estimated by using a normal kernel density; red lines represent 25th, 50th and 75th percentiles. Anomaly distributions significantly different from the climatological sample are displayed in green (p-values of the Kolmogorov-Smirnov test used to assess the significance of the differences are reported in Table S5). In titles, the number of available daily observations and the p-value of the ANOVA used to assess the WR characterisation are reported in brackets.

[Figure]

**Figure 11.** Environment to circulation characterisation: CAMS WR frequency anomaly for each quartile of the AOD anomalies at 500 nm at the AERONET stations. Values represent percentage changes relative to climatological frequencies. Green boxes highlight significant changes in the WR frequencies, i.e. frequency anomalies exceeding the critical threshold (11.07) for the chi-squared statistics with 5 degrees of freedom at 95% level of confidence (chi-squared statistics are reported in Table S6). In brackets, the number of available daily observations are indicated.

[Figure]

**Figure 12.** ERA5 WR frequency 1981-2020: interannual variability.

[Figure]

**Figure 13.** Interannual correlation over the period 1981-2020: SA+ and SA- frequency from the ERA5 classification vs (a, b) HadI sea surface temperature and (c, d) ERA5 geopotential height at 200 hPa in ASO. Shadings display significant correlations at 95% level of confidence. Time series are detrended and standardised.

---

## Author Response (AR1)

**General Comments**

The authors present a classification of the circulation over southern Africa and the South Atlantic into six weather regimes (WRs). They link these WRs with the transport of biomass burning aerosol (BBA) via the corresponding wind field and with the aerosol optical depth (AOD). Furthermore, they investigate whether there are significant links between the observed AOD at surface stations and the WRs.

This study appears to be the first linking WRs to aerosol transport in this region. It provides valuable insight in the circulation over the South Atlantic and southern Africa and shows significant relations between AOD at surface stations and some of the WRs. When WR classifications have been performed in the Northern Hemisphere, e.g. the Euro-Atlantic sector, these regimes are quite persistent and describe variability on timescales of 5-15 days, i.e. for a large part beyond synoptic timescales, while most WRs studied here describe precisely that synoptic variability in the region. Therefore, the regimes in this region are more a useful descriptive tool for the circulation than persistent features in phase space. Nevertheless, these WRs prove to be well suited for studying BBA transport in this region and could have a lot of potential for studying other circulation processes affecting southern Africa. The authors get this across quite well, although I think the potential of the WR can be emphasized more.

We thank the Reviewer for her/is interest in our work and the time s/he spent in making constructive comments, which helped us in substantially improving the quality of the manuscript. We provide below a point-by-point response to all the comments/requests. Reviewer's comments are in black, our responses are in blue.

**Specific Comments**

Can you comment on the robustness of the WRs? The time series used is relatively short, being only 15 years of ASO data, for the classification of WRs. It would be good to know the WRs are robust with respect to the reanalysis products used by computing the WRs for e.g. ERA-5 which has a longer available time pereiod.

Are the WRs you obtain dependent on the selected domain? Most of the variability, especially in WR1, 3, 4 and 5, is south of 20S. Do the WRs change if you e.g. only take the domain up to 10S, or extend the domain beyond 40S? A similar question on the number of PCs, have you looked into any dependence of the WRs on this?

We agree with the reviewer that an assessment of the robustness of the WR classification is needed. However we believe that the synoptic characterisation of the aerosol transport should be performed by using the CAMS product, which assure coherence between atmospheric circulation and aerosol data. Therefore we decide to keep using the CAMS classification to characterise the aerosol transport, and perform an additional assessment of its robustness following your suggestions. We tested: the sensitivity of the WR classification in CAMS to changes in the geographical domain and the number of retained PCs. Moreover the WR classification defined in 2003-2017 is compared with a WR classification defined in 1981-2020 by using ERA5 data. Section S2 has been added in the Supplement to discuss the sensitivity of the CAMS classification to changes in the geographical domain and the number of retained PCs, WR centroids and frequencies from the ERA5 classification have been added to Fig. 4 and 7, and a long paragraph has been added to Section 3.1 to discuss the classification in ERA5 and summarise the sensitivity tests. We can conclude that:

*"The sensitivity tests performed on the WR classification of the CAMS data show a high degree of robustness with respect to changes in the time period, and a good degree of robustness with respect to changes in the geographical domain and the retained variance, highlighting that the classification well represents the main features of the synoptic circulation in the region".*

You comment on the limited coverage of the CAMS reanalysis not being sufficient for defining a climatological seasonal cycle (p4, 100-102). As you only consider three months of data, would it make sense to simply use a fixed climatology, i.e. average over all months, with respect to which to compute the anomalies? And would this affect the WRs?

Removing the ASO climatology would be problematic because the emission of BBA from the source region (see the organic matter mixing ratio at 10m, averaged in the emission region in Tropical Africa, Figure R1), and consequent transport (see Fig. 1) largely vary from August to October, in terms of both intensity and variability. Therefore, the removal of a fixed climatology would result in biased daily anomalies, too high in August and too low in October. We believe that the definition of daily anomalies as the high frequency component of the day-by-day variability is well suited for studying the synoptic variability of both circulation and BBA transport in ASO.

[Figure]

**Figure R1.** Climatology of the organic matter mixing ratio at 10 m in the emission region in Tropical Africa, used to estimate the intensity of BBA emission.

In your study you use 6 WRs, and in the supplement you also show results for 7 WRs. Why did you choose to work with 6 WRs in the main manuscript instead of 7? If it's because the results simply are better, could you state that more clearly in the main manuscript. In the end that's the aim of using the WRs. From my point of view, selecting 6 only because it's the lowest significant number of WRs would already be a valid argument. Also, I'm quite surprised 2 WR yields such a poor classificability index (supplement), can you comment on this?

We first highlight that the red-noise test shows that the 6 and 7 class partitions are both significant and correspond to physically coherent atmospheric patterns, and partitioning the synoptic variability into 6 or 7 regimes would be equally appropriate. The choice of focusing on the 6 WR classification is mainly based on the fact that this is easy and convenient to interpret: it results in one high-pressure regime, one low-pressure regime (denoting contrasted intensities in the strength of the South

Atlantic/St. Helena High), and 4 regimes depicting transient disturbances in the middle latitudes, i.e. the propagation and life cycle of temperate waves embedded in midlatitude dynamics, and materializing the synoptic-scale variability that develops there. In addition, the comparison between the 6 and 7 class partitions shows that the 6 WR classification performs (slightly) better in characterising the BBA transport. This is now clearly stated at the end of Section 2.3:

*"The synoptic characterisation of the BBA transport is performed by using both the 6-class and the 7-class partitions. This study focuses on the 6-class partition, i.e. the classification with the lowest significant number of WRs, which leads to physically coherent atmospheric patterns describing the main features of the synoptic variability (see Section 3.1). Furthermore, the comparison between the 6 and 7 class partitions shows that the 6 WR classification performs better in characterising the BBA transport in the region (see Section S3)".*

Concerning the low classificability index associated with the 2-class partitioning, we do agree that in most cases it is much larger than here (e.g., Pohl et al. 2021, Supplementary Material, https://journals.ametsoc.org/view/journals/apme/aop/JAMC-D-20-0255.1/JAMC-D-20-0255.1.xml). A possible reason for this low score is that two regimes could not perform well to depict transient wave activity in the southern part of our domain. Yet, such transient disturbances are, by far, associated with the largest variance of the input field (geopotential height anomalies), attracting 4 regimes out of 6 to depict the eastward propagations of such synoptic disturbances. Consequently, these regimes are statistically instable and weakly coherent, because they are unable to depict well the dominant modes of variability in this region.

Could you expand on the computation of BBA emission and transport? For the transport, are you discussing anomalies with respect to the low frequency component (season-wise) or absolute values? Similarly, for AOD, are you plotting anomalies? It would be good to clarify throughout.

As stated in Section 2.1:

*"The BBA emission is estimated by the organic matter mixing ratio at 10m, the BBA transport is estimated as the product of organic matter mixing ratio and wind at 700 hPa, and the aerosol spatial distribution is represented by the aerosol optical depth (AOD) at 550 nm (Fig. 2a)".*

All the data from ERA5, CAMS and AERONET are used in the synoptic classification and characterisation (Fig. 4, 6, 9, 10 and 11) as daily anomalies, computed by removing the low frequency component, as stated in Section 2.1 and 2.2. The nature of the displayed variables is also stated in figure captions. In the revised version of the manuscript, in Section 2.3 we further specify that daily anomalies are used in the PCA:

*"The atmospheric circulation is first characterised by isolating the main modes of variability represented by the empirical orthogonal functions (EOFs) derived from a principal component analysis (PCA) of the **geopotential height daily anomalies**".*

In Section 2.4, we state that:

*"1) **Daily AOD anomalies** are linked to the corresponding WR and grouped, and statistical differences among groups are investigated (circulation-to-environment approach, C2E)".*

*"2) **Daily AOD anomalies** are divided into quartiles, and the changes in the WR occurrences within each quartile are studied (environment-to-circulation approach, E2C)".*

The WRs come in two types, with 2 and 6 describing the oscillation of the South Atlantic pressure field and the other four representing propagating disturbances along the midlatitude mean flow. The

latter four are not very persistent, and represent a travelling wavetrain (1-5-3-4). How do these two sets of WRs relate and could you also study the BBA transport using a travelling wave perspective?

Thank you for this comment, which prompted us to do some additional analysis adding robustness to our findings. Persistence is studied by analysing the length of the WR sequences, showing short persistence for both the SA and ML regimes (Fig. 5), and indicating synoptic variability as the dominant feature of the atmospheric variability in the region. A composite analysis from the occurrence of the WR (day 0) to day +4 shows that the SA regimes are characterised by pressure oscillations, with around 3 day lifetime, while the ML regimes represent a propagative pattern, with around 4 day lifetime (ML1 → ML2 → ML3 → ML4, see Fig. 6). This is now discussed in Section 3.1. The synoptic variability associated with the WR occurrence is also reflected in the BBA transport, as illustrated by a lead-lag correlation analysis of the CAMS AOD in the South Atlantic and southern Africa (Fig 9). The lead-lag correlation shows *"8 day period for the AOD anomalies to build up in the tropical South Atlantic along the easterly route from tropical Africa"*, and *"a 6 day period for the river of smoke to build up in the South Atlantic and move eastward across southern Africa"*, as discussed in Section 3.2.

You look into the sub-seasonal variability of the WR regime frequencies (sec 3.1, p7). I am unsure whether you have sufficient data to state that these differences are robust, as noise will affect these results. Specifically, if you have insufficient data to define a seasonal cycle for it is affected by interannual variability, do you have enough to do something similar for the WR frequencies? I.e. on average you have 15x30/6 = 75 days in each WR during one month, to which changes of 5-10 days in occurrence can have quite an impact.

We do agree with the Reviewer that there are not enough data in CAMS for assessing the intraseasonal variability of WR frequencies. In the new version of the paper, the seasonal frequency of CAMS WRs is presented in Fig. 7. In addition, we show in Fig. 7 the seasonal and intraseasonal frequencies of the ERA5 WRs, computed across 40 years, from 1981 to 2020:

*"The availability of 40 year time series in ERA5 allows to robustly estimate WR frequencies at the intraseasonal time scale (Fig. 7). Differences are limited to 1-2%, with the exception of ML2 and ML3, increasing by 3% and decreasing by 4% during the season, respectively".*

Can you expand the discussion of the link between your WR and the SAM, which you briefly mention (top of p8)? Why do you think WR6 relates better to the SAM than WR2? As WR2 overall occurs more often, it would be interesting to know any discrepancies. Can you include the results that are not shown in the supplement?

The analysis of the WR-SAM relationship has been revisited by using the long term ERA5 classification, and so the discussion of the results. The WR-SAM relationship has been analysed on daily and interannual time scales by comparing with the SAM daily and monthly indices provided by the NOAA Climate Prediction Centre (https://www.cpc.ncep.noaa.gov/products/precip/CWlink/daily_ao_index/aao/aao.shtml). The daily connection has been explored by means of the C2E and E2C approaches and a new Figure 8 has been added to the main text. The analysis of the daily variability in the period 1981-2020 now shows that more persistent WRs (SA+ and SA-) are associated with positive SAM phases, i.e. with weaker westerlies at midlatitudes. Conversely, the "travelling" WRs (ML1-4) are associated with negative SAM phases, i.e. with stronger westerlies. The discussion in Section 3.1 has been expanded.

The WR-SAM interannual relationship has been also investigated by correlating the 1981-2020 WR frequency with the SAM index and the 700 hPa geopotential height. No significant signal is detected

(see Fig. R2), as expected because of the synoptic variability dominating the WRs. A sentence has been added to Section 3.4:

*"The analysis of the WR-SAM relationship at the interannual time scale shows poor results when the WR frequency time series are correlated with the SAM monthly index. Similarly, the correlation between the WR frequency time series and the monthly averages of the geopotential height at 700 hPa in the Southern Hemisphere does not show evident correlation patterns (not shown)".*

[Figure]

**Figure R2.** ERA5 WR frequency vs SAM index, for ASO 1981-2020.

Would there be any effect of the duration of the WRs onto the CAMS AOD, and that of the surface stations? I can imagine that if a regime lasts longer the effect on AOD is stronger as well, more so for WR2 and 6. Also, is it possible that there is an effect of past WRs (i.e. delayed) onto the measured AOD, possibly linking to the discrepancies between surface station AOD values and those of CAMS?

The analysis of WR sequences shows short persistence, highlighting the dominance of the synoptic variability in the atmospheric and aerosol dynamics, and suggesting limited impact of persistence on AOD anomalies at surface stations. The transient pattern characterising the ML WRs suggests the possibility of predicting AOD anomalies with a 2-3 day lead time. However, the C2E characterisation of in situ observations shows little improvement for 1 and 2 day lead times (see Fig. R3 and R4), and ambiguous results for 4 day lead time (see Fig. R5). We agree that the predictive performance of the WR classification is worth of being further investigated, however the limited availability of in situ measurements prevents a robust skill assessment. Such a research could be done on data with larger time-space coverage (e.g. satellite products), but it is beyond the scope of this paper.

[Figure]

**Figure R3.** Circulation to environment characterisation: for each AERONET station (Ascension Island, Bonanza, Namibian Stations, Upington), (left panel) distribution of the AOD anomalies at 500 nm, and (right panel) for each CAMS WR, with 1 day lead time. Anomaly distributions significantly different from the climatological sample are displayed in green (p-values of the Kolmogorov-Smirnov test used to assess the significance of the differences are reported in Table S5). In titles, the number of available daily observations and the p-value of the ANOVA used to assess the WR characterisation are reported in brackets.

[Figure]

**Figure R4.** Same as Fig. R3, but for 2 day lead time.

[Figure]

**Figure R5.** Same as Fig. R3, but for 4 day lead time.

Three (maybe four) stations are located close together (GO, HB, HE, and BO(?)). Differences in AOD distributions (Fig 6) for each of the WR thus could be primarily caused by local effects, e.g. for WR3, can you comment on this? Similarly for the WR frequency anomalies in Fig 7. Also, would it make sense to pool these stations together to get more robust statistical results (I am not familiar with the data itself, so do not know whether this would be sensible)?

Thank you for the suggestion. Indeed, the three stations in western Namibia are characterised by slightly different local conditions, which could add local effects to the AOD measurements. Namely, Henties Bay is a coastal site, exposed to both marine and mineral dust aerosols; Gobabeb is in the Namib Desert, exposed to mineral dust aerosols; HESS is located inland in the savannah, exposed to possible local sources of BBA. Therefore, to filter out possible local effects, we built a time series associated with the three stations (referred as Namibian Stations, NS) computed as the average AOD when observations are available in at least 2 stations out of 3. This also allows to have a time series longer than the three individuals (276 observations, spanning from 2013 to 2017). The combined time series is presented in Section 2.2, the analysis is added to Fig. 3, 10 and 11, and results are discussed in the main text as representative of mean conditions in western Namibia (Section 3.3).

More generally, station data is limited. Can you comment on how many days are in each of the WR for the different stations? This would help the reader get an idea of the robustness of the results in Fig 6 and 7.

The number of available observations for each station is reported in Table 1 and the number of data points used in the analysis is displayed in Fig. 3, 10 and 11. In Section 2.2 we highlight that stations with less than 100 observations are discarded:

*"Among the available stations, St. Helena (15.9°S, 5.7°W) and Wits University (26.2°S, 28.0°E) are not included because of the limited coverage (less than 100 observations during the study period)"*.

In the revised version, a sentence has been added at the beginning of Section 3.3 to highlight this aspect:

*"It is highlighted that data availability and coverage in most of the stations is limited (see Table 1), resulting in circa 20-40 observations per WR on average. Only the station in Ascension Island covers the whole period analysed, providing more than 600 observations, i.e. circa 100 observations per WR on average"*.

In the Conclusions we also state that:

*"Limited data availability in most of the stations prevents a robust statistical validation of the synoptic characterisation of observations at the regional scale"*.

How robust are your results on interannual variability, having only ~90 days with associated regimes for each year? In my experience interannual variability in regime frequencies of reanalysis products can be very large, and does not necessarily allow for clearly showing a signal through the noise. Could you use a longer reanalysis to look into links with e.g. ENSO to make these links more robust? I am not surprised WR 1,3,4,5 do not show any significant links with SST, as these are short-lived regimes. Have you thought about the persistence effect of the regimes and whether that could be affected (I appreciate there's too little data for this)?

Thank you for this comment. After building the WR classification in ERA5, which results being consistent with the analysis of CAMS data, in the revised version of the manuscript we use this longer time series for analysing the interannual variability across 40 years. New results show no large

differences among the WRs, with no significant trends and comparable variability (STD between 3% ad 4%). We highlight that the WR classification shows dominant synoptic variability and short persistence, therefore we expect no strong signal at the interannual time scales (e.g. there is no relationship with the interannual SAM). When the WR-SST teleconnection is explored, a weak relationship is found between SA regimes (showing some persistence) and SST anomalies in the tropical Pacific, along with a possible teleconnection pattern (see Fig. 13 and Section S4). The analysis of the WR-AOD interannual relationship has been suppressed because of the limited AOD data coverage, in both the reanalysis products and in situ observations.

Most plots are not very clear and take a long time looking at to understand what they show. Could you add labels to all axes and colorbars where they are not there, increase the fontsize of ticks, labels and captions, and remove any redundant information. The z700 patterns of the regimes are not clear at the moment (Fig 4), maybe increasing the line thickness would help, or otherwise I think it would be good to add them in a separate figure, as they are important to get across well. For Fig 6 and 7 it would be good to clearly see which results are significant and which are not, e.g. change the colour in Fig 6 and lines around the relevant cells in Fig 7. Moreover, there is some discrepancy between the p-values given in Fig 6 and Tab 3. I think it is sufficient to show these values only in the table if you indicate the significant ones in another way.

The quality of the figures has been improved. Circulation patterns are shown by thicker contours and WRs are also shown in separate figures without superimposed BBA (see Fig. S4 and S7). Fig. 10 and 11 have been modified, highlighting the significant differences. Tables with ANOVA, Kolmogorov-Smirnov and chi-squared statistics have been moved to Section S5 in the Supplement.

**Technical Corrections**

Could you rearrange the WRs such that their order links better to the regimes themselves? For example, change WR2, 6 to WR1 and 2, and WR1, 5, 3, 4 to WR3, 4, 5, 6 (in order of transitions), respectively. It would also be good to line up the 7 WRs in the supplement with the six in the manuscript.

Thank you for this useful suggestion. The WRs have been rearranged and labelled accordingly to their circulation features: SA+ and SA- for the pressure anomalies in the South Atlantic, ML1-4 for the midlatitude pressure anomalies. The ML regimes are labelled to highlight the propagative nature of the transition, with pressure anomalies in ML1 moving eastward into ML2, 3, and 4. The 7 class WR classification has been labelled following the same principle, and the WRs from both the classifications are now displayed in the same figure in the Supplement (Figure S7 in Section S3).

Please check the figure numbers as they are not everywhere correct, e.g. p9, 268, Fig. 6a/6b should be Fig 7a/7b and p11, 318, Fig. 10 should be Fig. 11. I might have missed some others.

Figures in the main text have been reorganised, and new figures have been added. The text-figure correspondence has been checked and fixed where necessary.

Throughout the manuscript: South Atlantic -> the South Atlantic

This has been corrected where necessary, thank you.

p2, 27 by -> of

Corrected.

p2, 35-39 Partly repeating what is mentioned earlier in the abstract, I would recommend rewriting or weaving it into the earlier part of the abstract

The last paragraph of the abstract has been rephrased:

*"The skill in characterising the BBA transport shown by the WR classification indicates the potential for using it as a diagnostic/predictive tool for the aerosol dynamics, which is a key component for the full understanding and modelling of the complex radiation-aerosol-cloud interactions controlling the atmospheric radiative budget in the region."*

p3, 44 Remove "on"

Removed.

p3, 47 originated -> originating

Changed.

p3, 48 Insert "and is" between "(Fig. 1a)" and "a prominent"

Done.

p3, 52 definition -> term

Changed.

p3, 59 shed light -> shed light on

Corrected.

p3, 63 the Antarctica -> Antarctica

Corrected.

p4, 76-78 This sentence is nearly the same as the one above

Yes, the sentence has been removed.

p5, 106 You already mention AOD here as an abbreviation but only define it in line 114, also it's already mentioned in the caption of Fig 1, so might be good to clarify it there as well

The acronym is now defined here and in the caption of Fig. 1.

p5, 124 with -> of

Corrected.

p5, 131 as linear -> to be linear

Changed.

p6, 139-140 before to perform the linear regression -> before the linear regression is performed

Changed.

p6, 148 to mask -> mask

Corrected.

p7, 182 southward the midlatitude westerly flow -> the midlatitude westerly flow southward

Changed.

p7, 182-187 Do you have any references supporting this, or is it solely based on Fig 2a?

This is basically the description on Fig. 2a, which represents the climatology for the ASO season. A reference to Adebiyi and Zuidema (2016), describing the atmospheric circulation and aerosol climatology in September-October, has been added.

p7, 189 filed -> field

Corrected.

p7, 192 the WR2 -> WR2

The sentence has been rephrased.

p7, 203 Can you change the order of the preferred transitions in the brackets, so they are in order of transitioning?

This sentence has been rephrased, and the WR transitions are organised to highlight the eastward propagative pattern: ML1 → ML2 → ML3 → ML4.

p8, 215 coherently -> coherent

The sentence has been removed.

p8, 216 results statistically -> results are statistically

The sentence has been removed.

p9, 249 Can you repeat the null hypothesis here?

Done.

p12, 357 WR then clustering -> WR clustering

Corrected.

**Anonymous Referee #2**

This paper presents a statistical analysis of various climate fields to relate the tropospheric air quality in the Southern African continent, atmospheric circulation and SST.

The general idea is interesting and the paper is well structured (although I appreciate when figures appear in the text, where they are cited, rather than at the end of the manuscript, which makes reading rather tedious on a computer).

We thank the Reviewer for her/is interest in our work and the time s/he spent in making constructive comments, which helped us in substantially improving the quality of the manuscript. We apologise for the manuscript layout, but we have many figures with many panels and this is not easily manageable by MS Word. We provide below a point-by-point response to all the comments/requests. Reviewer's comments are in black, our responses are in blue.

My major comment is on the application of the statistical methodology. The authors seem to use ~15 years of geopotential data from the CAMS reanalysis. The rationale is that AOD data are only available on that period. But the authors use SST data that cover more than one century (and use only a small subset). I think it would be more appropriate to apply the k-means algorithm on a longer period of time (e.g. with ERA-I, ERA5, or NCEP reanalyses) to compute weather regimes in a statistically robust way, and then classify CAMS data onto such weather regimes. This would reduce the uncertainty on the computation of WRs.

We agree with the reviewer that an assessment of the robustness of the WR classification is needed. However we believe that the synoptic characterisation of the aerosol transport should be performed by using the CAMS product, which assure coherence between atmospheric circulation and aerosol data. Therefore we decided to keep using the CAMS classification to characterise the aerosol transport, and assess its robustness as you (and Referee #1) suggest. We tested: the sensitivity of the WR classification in CAMS to changes in the geographical domain and the number of retained PCs. Moreover the WR classification defined in 2003-2017 is compared with a WR classification defined in 1981-2020 by using ERA5 data. Section S2 has been added in the Supplement to discuss the sensitivity of the CAMS classification to changes in the geographical domain and the number of retained PCs, WR centroids and frequencies from the ERA5 classification have been added to Fig. 4 and 7, and a long paragraph has been added to Section 3.1 to discuss the classification in ERA5 and summarise the sensitivity tests. We can conclude that:

*"The sensitivity tests performed on the WR classification of the CAMS data show a high degree of robustness with respect to changes in the time period, and a good degree of robustness with respect to changes in the geographical domain and the retained variance, highlighting that the classification well represents the main features of the synoptic circulation in the region".*

The ERA5 classification is used in the revised version of the manuscript for the analysis of the intraseasonal and interannual variability of the WRs in the period 1981-2020 (see Fig. 7, 8, 12 and 13).

My second methodological suggestion is to use a cross-validation approach to E2C and C2E, by "learning" the associations between WR and AOD on a decade, and "testing/validating" this association on the remaining 5 years. This would give credence to the alleged predicting power of the statistical approach.

Thank you for this suggestion. We do agree that the predictive potential of the WR classification is worth to be investigated. However, the shortness of the AERONET time series prevent the

application of the suggested cross validation approach. Please note that only in Ascension Island observations cover the whole studied period (2003-2017) with only 612 data points (40 per year on average), while the continental stations only cover 2-4 years, with at best less than 300 data points (see Table 1). However, we have tested the C2E approach with different time leads, namely 1, 2, and 4 days (see Fig. R1, R2 and R3). Results show little improvement for 1 and 2 day lead times, reflecting the 2-3 day persistence of the WRs, and ambiguous results for 4 day lead time. Predictive skill assessment of the WR classification should be done on data with larger time-space coverage (e.g. satellite products), but it is beyond the scope of this paper.

[Figure]

**Figure R1.** Circulation to environment characterisation: for each AERONET station (Ascension Island, Bonanza, Namibian Stations, Upington), (left panel) distribution of the AOD anomalies at 500 nm, and (right panel) for each CAMS WR, with 1 day lead time. Anomaly distributions significantly different from the climatological sample are displayed in green (p-values of the Kolmogorov-Smirnov test used to assess the significance of the differences are reported in Table S5). In titles, the number of available daily observations and the p-value of the ANOVA used to assess the WR characterisation are reported in brackets.

[Figure]

**Figure R2.** Same as Fig. R1, but for 2 day lead time.

[Figure]

**Figure R3.** Same as Fig. R1, but for 4 day lead time.

**Minor comments**

The first paragraph of the introduction states that aerosols modify the radiative properties of the atmosphere. Fine. "As a consequence, they can influence on the atmospheric synoptic and large-scale dynamics" seems strange, as the radiative properties of aerosols are rather local, which contradicts large-scale atmospheric motion, where radiation is not so crucial. Please explain.

Thank you for this comment, the sentence is actually misleading. We referred to the role of aerosols as climate forcing, and we made the equivalence between climate and large scale dynamics, which is, we do agree, not appropriate. The sentence now reads:

*"As a consequence, aerosols can influence the atmospheric and climate dynamics"*.

The end of the introduction lacks a paragraph that states the scientific question that the manuscript is dealing with. At present, the introduction states rather general questions, then states what the authors intend to do. How this endeavor corresponds to the many general questions seems to be left to the imagination of the reader.

The specific scientific questions we aim to respond are formulated explicitly in the Introduction:

*"The scope of this paper is to fill the gaps in the understanding of atmospheric and aerosol dynamics during austral winter in the southern Africa/South Atlantic sector, by providing a characterisation of the synoptic variability of the atmospheric circulation, and determining the circulation patterns controlling the transport of BBA from the tropics to the extratropics. To this aim, an objective weather regime (WR) classification…"*

When the authors compute the correlation between SST and WR frequency (Figure 9), they could do this on a much longer period, as WR can be determined from longer reanalyses. This would provide a more robust assessment of interannual relations.

Thank you for this comment. After building the WR classification in ERA5, which results to be consistent with the analysis of CAMS data, in the revised version of the manuscript we use this longer time series for analysing the interannual variability across 40 years (Fig. 12). New results show no large differences among the WRs, with no significant trends and comparable variability (STD between 3% ad 4%). We then use the interannual frequencies to investigate possible teleconnections. As expected, the correlation between SST anomalies in the tropical Pacific and SA+/SA- is limited when a longer time span is considered. However, the analysis of the associated

atmospheric pattern shows a possible teleconnection (see Fig. 13). We highlight that the WR classification show that synoptic variability is dominated by transient disturbances, and the role of interannual teleconnections is limited.

Could the authors compare their results with computations of particle trajectories, for well-chosen events?

We do agree that computing particle trajectories for selected events would provide an interesting comparison with our results. However, the scope of the paper is to provide a comprehensive picture of the synoptic variability of circulation and aerosols, rather than investigating single events. Therefore, we applied a lead-lag correlation analysis to the CAMS AOD daily anomalies to highlight the development of the BBA transport events in the South Atlantic and southern Africa in the context of a wave pattern dominating the synoptic variability in the region (see Fig. 9 in the revised version). Results show that the AOD anomalies are modulated at the same pace of the WR lifetimes (3 days for the SA regimes, 4-5 days for the ML regimes), confirming the WR control on the BBA transport.

The paper does not present any discussion of comparisons with already existing results. I am not an expert on the subject, but I would have expected that the results reported by the authors could be placed in a context of existing literature.

Thank you for this comment, we agree that our results need to be placed in the context of existing research. As pointed out in the introduction, in our knowledge, no long term characterisation of the BBA transport in the South Atlantic/southern Africa has been presented in the literature to date. For this reason, comparison is only possible with papers analysing short time periods or case-studies. In the revised version of the manuscript, we discuss our findings in comparison with results from the SAFARI-92 and SAFARI 2000 campaigns. A new paragraph has been added in the conclusions:

*"The analysis of the regional circulation patterns controlling the BBA transport the South Atlantic/southern Africa sector is reported in literature mainly as a complement in the discussion of field campaign results. During the SAFARI-92 field experiment, Lindesay et al. (1996) reported pronounced BBA transport across southern Africa towards the Indian Ocean, in association with El Niño conditions and intensified continental high. Conversely, during the SAFARI 2000 campaign (Swap et al., 2003), Stein et al. (2003) found an association between the occurrence of rivers of smoke heading towards the Indian Ocean and increased westerly waves and weaker continental high, concomitant with La Niña conditions (see also Garstang et al., 1996). These contrasting conclusions likely originate from to the limited robustness of the analysis due to the shortness of the observation periods. Based on a longer dataset, the WR characterisation suggest a key role of the westerly waves in controlling the rivers of smoke, supporting the hypothesis of Garstang et al. (1996), although it remains inconclusive concerning the role of ENSO phases".*